# Scleraxis-lineage cell depletion improves tendon healing and disrupts adult tendon homeostasis

Katherine T Best[1], Antonion Korcari[1,2], Keshia E Mora[1,2], Anne EC Nichols[1], Samantha N Muscat[1], Emma Knapp[1], Mark R Buckley[1,2], Alayna E Loiselle[1,2]*

[1]Center for Musculoskeletal Research, University of Rochester Medical Center, Rochester, United States; [2]Department of Biomedical Engineering, University of Rochester, New York, United States

**Abstract** Despite the requirement for *Scleraxis*-lineage (Scx[Lin]) cells during tendon development, the function of Scx[Lin] cells during adult tendon repair, post-natal growth, and adult homeostasis have not been defined. Therefore, we inducibly depleted Scx[Lin] cells (ScxLin[DTR]) prior to tendon injury and repair surgery and hypothesized that ScxLin[DTR] mice would exhibit functionally deficient healing compared to wild-type littermates. Surprisingly, depletion of Scx[Lin] cells resulted in increased biomechanical properties without impairments in gliding function at 28 days post-repair, indicative of regeneration. RNA sequencing of day 28 post-repair tendons highlighted differences in matrix-related genes, cell motility, cytoskeletal organization, and metabolism. We also utilized ScxLin[DTR] mice to define the effects on post-natal tendon growth and adult tendon homeostasis and discovered that adult Scx[Lin] cell depletion resulted in altered tendon collagen fibril diameter, density, and dispersion. Collectively, these findings enhance our fundamental understanding of tendon cell localization, function, and fate during healing, growth, and homeostasis.

**\*For correspondence:**
alayna_loiselle@urmc.rochester.edu

**Competing interests:** The authors declare that no competing interests exist.

## Introduction

Despite the significant efforts toward improving tendon healing and regeneration, the specific cellular contributions during tendon healing have not been extensively characterized (*Nichols et al., 2019*). While many studies have examined the potential of using various stem cell populations to promote healing, originating from both tendon intrinsic (*Walia and Huang, 2019*) and extrinsic sources (*Costa-Almeida et al., 2019*), little focus has been directed toward defining the functions and therapeutic potential of tendon cells during tendon healing following an acute injury. Tendon cells are increasingly recognized as a heterogenous population of cells where many, but not all, express the gene *Scleraxis* (*Scx*) (*Best and Loiselle, 2019*; *Kendal et al., 2020*; *De Micheli et al., 2020*). Understanding the localization and function of tendon cell subpopulations during healing is likely to be instrumental in better defining the mechanisms that promote scar-mediated healing, which results in poor patient outcomes, and could therefore be used to develop pro-regenerative approaches to improve healing.

*Scx*, a basic helix-loop-helix transcription factor, is currently the most well-characterized marker that the field possesses to study tendon (*Schweitzer et al., 2001*). *Scx* has been utilized to examine tendon biology and development (*Murchison et al., 2007*; *Pryce et al., 2009*; *Huang et al., 2019*), healing and regeneration (*Sakabe et al., 2018*; *Howell et al., 2017*; *Best and Loiselle, 2019*; *Dyment et al., 2014*), differentiation (*Bavin et al., 2017*; *Chen et al., 2012*; *Alberton et al., 2012*; *Nichols et al., 2018a*), and mechano-transduction (*Maeda et al., 2011*; *Nichols et al., 2018b*). Functionally, *Scx* expression can drive matrix production and remodeling (*Sakabe et al., 2018*; *Leéjard et al., 2007*; *Levay et al., 2008*), epithelial-to-mesenchymal transition (*Al-Hattab et al.,*

2018), development of force-transmitting tendons (*Murchison et al., 2007*), tendon growth (*Gumucio et al., 2020*), and effect focal adhesion morphology (*Nichols et al., 2018b*). Despite the effort to understand the functions of *Scx* as a transcription factor, the function and requirement of *Scx*-lineage (Scx$^{Lin}$) cells in tendon repair, post-natal growth, and homeostasis has not been defined.

Previous studies examining the localization and role of tendon cells during healing are limited. *Scx*-GFP mice (*Pryce et al., 2007*) have been used to visualize tendon cells in many studies; however, previous work has suggested that extrinsic, paratenon-derived *Scx*-GFP⁻ cells can turn on *Scx* expression and become *Scx*-GFP⁺ by 14 days in a Patellar tendon injury model (*Dyment et al., 2014*). This makes interpretation using *Scx*-GFP mice complicated during in vivo studies as it becomes difficult to determine if a *Scx*-GFP⁺ cell is tendon-derived or simply activating *Scx* expression post-injury. Other studies have utilized the inducible Scx-Cre$^{ERT2}$ mouse model to label *Scx*⁺ cells prior to injury to allow tracking of these cells post-injury (Scx$^{Ai9}$). Howell et al. determined that while *Scx*-GFP cells and Scx$^{Ai9}$ tendon cells localized to the regenerated tendon in neonatal mice, these cells were not recruited to the scar tissue/bridging tissue during adult healing in Achilles tendon (*Howell et al., 2017*). In contrast, we have previously demonstrated that Scx$^{Ai9}$ tendon cells organize into a linear, cellular bridge spanning the scar tissue between the tendon stubs following acute injury and repair of the adult flexor digitorum longus (FDL) tendon (*Best and Loiselle, 2019*). While these studies suggest that tendon type (ex. flexor, Achilles, etc.) and injury model-specific (ex. transection with no repair, transection with repair, etc.) differences may modulate the Scx$^{Lin}$ cell contribution to injury, it also highlights the near complete lack of characterization of Scx$^{Lin}$ cell function in the healing process.

In the present study, we hypothesized that Scx$^{Lin}$ tendon cells would be required for successful healing in an adult model of acute flexor tendon repair by driving formation of a collagenous tissue bridge. We utilized a genetic mouse model of Scx$^{Lin}$ cell depletion to directly assess the function of tendon cells during healing and surprisingly discovered that depletion of Scx$^{Lin}$ tendon cells resulted in improved tendon biomechanical properties. We also examined alterations in wound healing-related cell populations, transcriptomics via RNA sequencing, and the effects of Scx$^{Lin}$ cell depletion on tendon post-natal growth and homeostasis.

## Results

### Successful ablation of *Scleraxis*-lineage tendon cells using diphtheria toxin receptor mouse model

To determine the feasibility of depleting tendon cells using Scx-Cre; Rosa-DTR$^{LSL}$ (ScxLin$^{DTR}$) mice (*Figure 1A*), diphtheria toxin (DT) was administered into the right hind paw for 5 consecutive days. Ten days after the final DT injection, both the injected and contralateral control hind paws were harvested (*Figure 1B*). This approach resulted in 57% depletion of tendon cells in uninjured ScxLin$^{DTR}$ FDL tendons relative to WT control littermates (p<0.0001) (*Figure 1C and D*). Tendon cell number was unaffected in the contralateral FDL, indicating that local DT injections did not induce cell death in locations other than the targeted hind paw (*Figure 1C and D*).

To better understand how this depletion regimen specifically effects Scx$^{Lin}$ cells we used Scx-Cre; Rosa-DTA$^{LSL}$; Rosa-Ai9 reporter mice (ScxLin$^{Ai9DTR}$) (*Figure 1E*). Samples were harvested at 24 and 38 days post-depletion (*Figure 1F*). At 24 days post-depletion, there was a 58.9% reduction in total tendon cells in uninjured ScxLin$^{Ai9DTR}$ FDL tendons relative to WT control littermates (p<0.0105) and a 68.48% depletion of Scx$^{LinAi9}$ cells in ScxLin$^{Ai9DTR}$ FDL tendons relative to WT ScxLin$^{Ai9}$ controls (p<0.0068) (*Figure 1G and H*). At 38 days post-depletion, total tendon cells were reduced by 60.42% (p<0.0091), and Scx$^{LinAi9}$ cells were reduced by 73.65% relative to ScxLin$^{Ai9}$ WT controls (p<0.0045) (*Figure 1G and H*). No significant differences in total tendon cell depletion efficiency (p<0.9935) and Scx$^{LinAi9}$ cell depletion efficiency (p<0.9359) were found between D24 and D38 post-depletion timepoints.

To understand how ScxLin$^{DTR}$ affected previously established tendon cell sub-populations (*Best and Loiselle, 2019*), we evaluated active Scx and S100a4 expression in ScxLin$^{DTR}$ and WT uninjured flexor tendons 10 days following the final DT injection. The number of Scx+ cells was significantly reduced in the ScxLin$^{DTR}$ tendons relative to WT littermates, as expected (p=0.0448) (*Figure 1—figure supplement 1A &B*). Similarly, when Scx+ cells were normalized to total cell

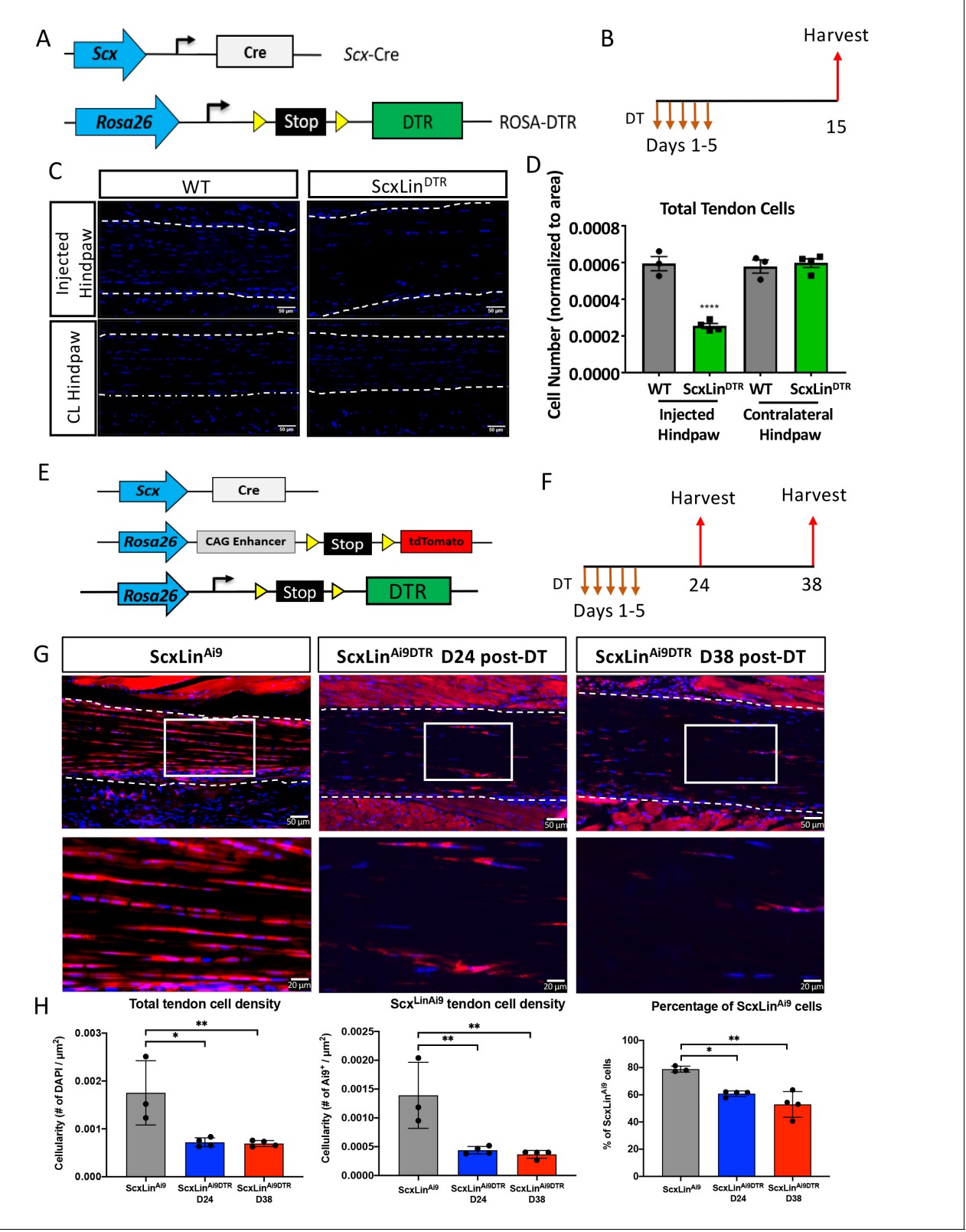

**Figure 1.** Efficiency of tendon cell and Scx[Lin] cell depletion. (A) To deplete Scx[Lin] cells, Scx-Cre mice were crossed to the diphtheria toxin receptor mouse (ScxLin[DTR]). (B) Mice received five hind paw injections of DT and were harvested 10 days after the final injection. (C) Sections from injected and contralateral (CL) hind paws from WT and ScxLin[DTR] mice were stained with DAPI, and total DAPI+ cells within the tendon (white outline) were quantified (D). (E) To determine the depletion efficiency specifically of Scx[Lin] cells, Scx-Cre; Rosa-DTR[LSL]; Rosa-Ai9 and Scx-Cre; Rosa-Ai9 reporter mice

*Figure 1 continued on next page*

*Figure 1 continued*

were given local, daily DT injections for 5 consecutive days and hind paws were harvested 24 and 38 days after the last injection (**F**) These are the contralateral control tendons from the mice in *Figure 4* that underwent tendon injury and repair. (**G**) Hind paws from ScxLin^Ai9 and ScxLin^Ai9DTR were probed for Red Fluorescent Protein (RFP; Ai9) expression and counterstained with the nuclear dye DAPI. (**H**) Total tendon cell density (DAPI⁺), total ScxLin^Ai9+ cell density and the percentage of ScxLin^Ai9 cells (ScxLin^Ai9+ cells/ DAPI⁺ cells) were quantified in ScxLin^Ai9 and ScxLin^Ai9DTR tendons and demonstrate a significant reduction of ScxLin^Ai9 cells in ScxLin^Ai9DTR relative to ScxLin^Ai9 WT controls. N = 3–4 per genotype. Two-way ANOVA with Sidak's multiple comparisons test used to assess statistical significance of tendon cell ablation between hind paw (injected with DT or contralateral) and genotype (ScxLin^Ai9 and ScxLin^Ai9DTR at 24 and 38 days). * indicates p<0.05 for the indicated comparison, ** indicates p<0.01 for indicated comparison, **** indicates p<0.0001 relative to all other groups.

The online version of this article includes the following figure supplement(s) for figure 1:

**Figure supplement 1.** *Scx+* and *S100a4+* tendon cells following Scx^Lin depletion.

**Figure supplement 2.** ScxLin^DTR does not cause substantial effects on surrounding tissue.

count, there was a trending reduction of Scx+ cells in ScxLin^DTR tendons compared to WT (p=0.1000) (*Figure 1—figure supplement 1C*). ScxLin^DTR tendons had a trending decrease in S100a4+ cells relative to WT littermates (p=0.0941) (*Figure 1—figure supplement 1D &E*). Interestingly, when S100a4+ cells were normalized to total cell count, there was no significant or trending difference between groups (p=0.3525), suggesting some remaining tendon cells may begin expressing S100a4 following Scx^Lin cell depletion (*Figure 1—figure supplement 1F*). ScxLin^DTR tendons exhibited a small number of apoptotic cells peripheral to, but not within, the tendon suggesting that a 10-day DT washout period was sufficient for all tendon-specific DT-induced cell death to occur (*Figure 1—figure supplement 2A*). Both ScxLin^DTR and WT littermates exhibited PCNA+ cells within the muscle, but not within then tendon, and ScxLin^DTR exhibited a few PCNA+ cells within the tendon that were not present in WT (*Figure 1—figure supplement 2B*). ScxLin^DTR tendons exhibited more F4/80+ macrophages peripheral to the tendon, potentially due to macrophage recruitment to the tendon to clean up apoptotic tendon cell debris (*Figure 1—figure supplement 2C*).

## Ablation of Scleraxis-lineage cells results in significantly increased biomechanical properties by day 28 post repair while not affecting gliding function

To define the functional effects of Scx^Lin cell depletion on tendon healing, mice received five local DT injections to deplete Scx^Lin cells followed by FDL repair 10 days following the final injection (*Figure 2A*). A trending improvement in MTP Range of Motion (ROM) in ScxLin^DTR repairs was observed at day 14, relative to WT littermates (WT vs ScxLin^DTR, p=0.0711), but this trend was absent by day 28 post-repair (*Figure 2B*). ScxLin^DTR healing tendons did not significantly differ in gliding resistance at either day 14 or 28 post-repair (*Figure 2C*). While biomechanical properties were not altered between groups at day 14, both stiffness and maximum load at failure were significantly increased in ScxLin^DTR healing tendons relative to wildtype littermates at day 28 post-repair (Stiffness: WT: 6.48 ± 0.75, ScxLin^DTR: 11.22 ± 1.83, p=0.0237; Maximum load at failure: WT: 1.54 ± 0.17, ScxLin^DTR: 2.44 ± 0.24, p=0.0061) (*Figure 2D and E*). Between days 14 and 28, WT tendon stiffness increased by 39.06% while ScxLin^DTR stiffness increased by 109.33%, and WT tendon maximum load at failure increased by 52.48% while ScxLin^DTR maximum load at failure increased by 162.37%, indicating that ScxLin^DTR repairs heal at an accelerated rate relative to wild-type littermates.

## Scleraxis-lineage cells are not required for the formation of a bridging collagen matrix during tendon healing

We have previously demonstrated that a cellular bridge corresponds to a region of bridging collagen matrix in the scar tissue (*Best and Loiselle, 2019*). While there were no apparent differences in tissue morphology between groups at days 14 or 28 post-repair (*Figure 3B*), we wanted to determine if tendon cell depletion prevented formation of the collagen bridge. Masson's trichrome staining revealed presence of bridging collagen through the scar in both groups at days 14 and 28, indicating that depletion of Scx^Lin tendon cells does not prevent formation of the bridging collagen matrix (*Figure 3C*).

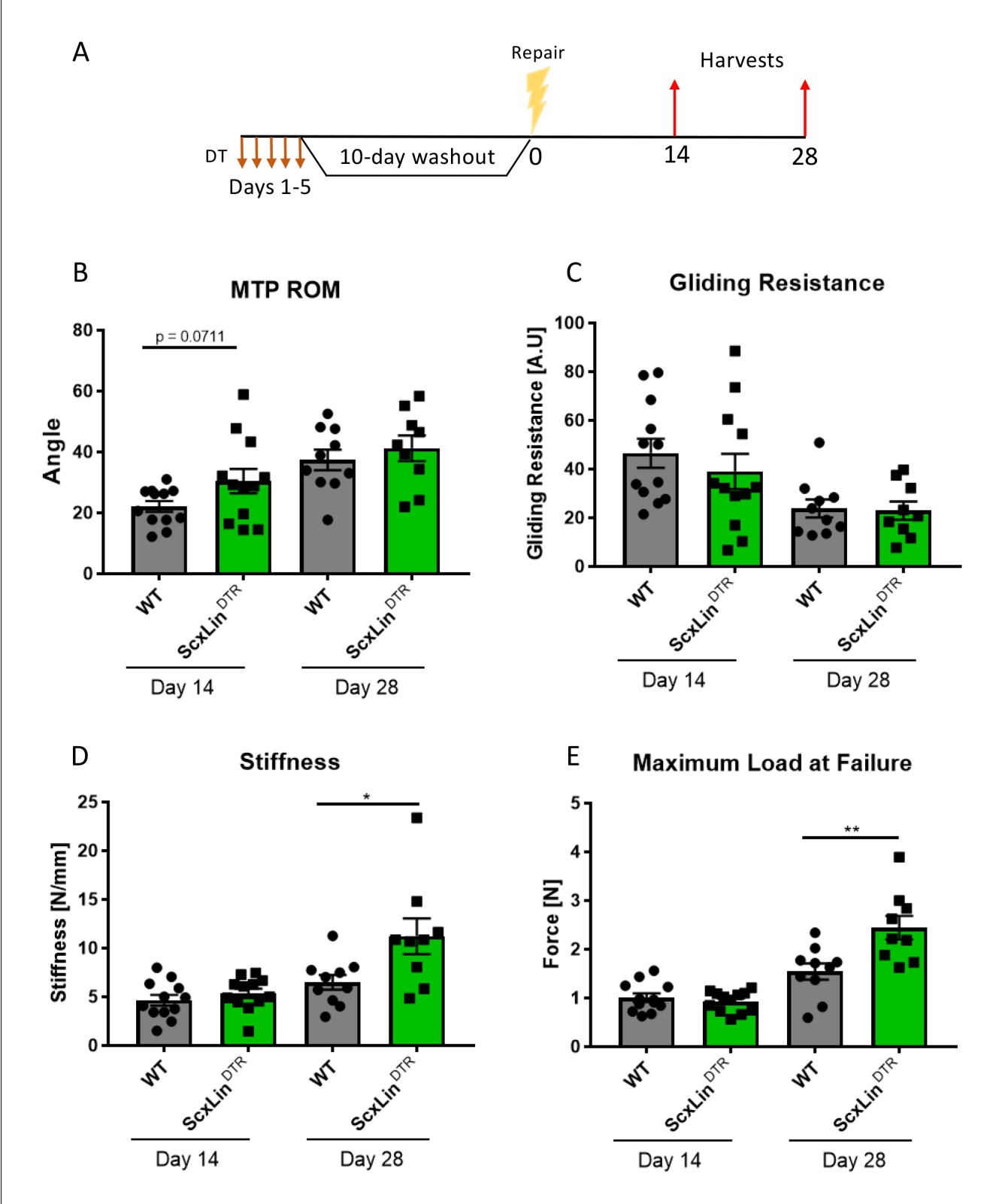

**Figure 2.** ScxLin[DTR] tendons heal with significantly increased biomechanical properties. Mice received five hind paw injections of DT on consecutive days, underwent flexor tendon repair surgery 10 days after the final DT injection, and were harvested at 14- and 28 days post-repair (**A**). Measurement of metatarsophalangeal (MTP) joint flexion angle (**B**), gliding resistance (**C**), stiffness (**D**), and maximum load at failure (**E**) of WT and ScxLin[DTR] repaired

*Figure 2 continued on next page*

*Figure 2 continued*

tendons. N = 9–12 per genotype per timepoint. Students t-test used to assess statistical significance between genotypes at a given timepoint. * indicative of p<0.05, ** indicative of p<0.01.

## $Scx^{Lin}$ cell depletion results in a significant lower $Scx^{lin+}$ cells during tendon healing

To better understand how depletion of $Scx^{Lin}$ cells prior to injury and repair affected $Scx^{Lin}$ cell density during healing, we traced $Scx^{LinAi9}$ cells at D14 and D28 post-injury (*Figure 4A*) No significant differences in $Scx^{LinAi9}$ cells were detected between WT ($ScxLin^{Ai9}$) and $ScxLin^{Ai9DTR}$ (p=0.3115) at D14 post-surgery (*Figure 4B,C*). In contrast, at D28 post-surgery, a significant decrease in $Scx^{LinAi9}$ cells was observed in $ScxLin^{Ai9DTR}$ repairs (p<0.0034) relative to WT repairs (*Figure 4B,D*). Collectively, these data suggest that depletion of $ScxLin^{Ai9}$ cells prior to injury does not alter the overall $ScxLin^{Ai9}$ content at D14, possibly due to additional labeling of cells that express *Scx* following injury. In contrast, by D28 the effects of depleting $ScxLin^{Ai9}$ cells prior to tendon injury is manifested in changes in both $ScxLin^{Ai9}$ content (*Figure 4B,D,E*) and phenotypic differences (*Figure 2*).

## $Scx^{Lin}$ depletion enhances myofibroblast content during tendon healing

We have previously demonstrated that elevated F4/80+ macrophages and αSMA+ myofibroblasts are associated with increased tendon maximum load at failure (*Best et al., 2019*). No significant differences in F4/80+ macrophages were detected between genotypes at either day 14 or 28 post-repair (*Figure 5B*). While αSMA+ myofibroblasts were not significantly altered at day 14 (D14: WT vs DTR, p=0.3790), $ScxLin^{DTR}$ healing tendons had significantly increased levels of αSMA+ myofibroblasts relative to wildtype littermates at D28 (p=0.0188) (*Figure 5C*). We have previously demonstrated that S100a4 is an important molecule that can influence tendon biomechanical properties and gliding function during healing (*Ackerman et al., 2019*). No differences in S100a4+ cells were observed in $ScxLin^{DTR}$, relative to WT at either day 14 or 28 post-repair (*Figure 5D*). Thus, αSMA+ myofibroblasts are the most likely candidate driving the increased biomechanical properties seen in $ScxLin^{DTR}$ healing tendon, of the cell populations investigated, consistent with their roles in with matrix deposition, organization, and contraction. To demonstrate both the specificity of the αSMA+ myofibroblast response to injury and that this tendon injury and repair model does not induce degeneration of the proximal/ distal ends of the tendon due to altered loading, we examined αSMA expression adjacent to the repair site. No αSMA staining was observed in the proximal/distal native tendon away from the repair site at D14 or D28 (*Figure 5—figure supplement 1*).

## Identification of differentially expressed genes following $Scx^{Lin}$ depletion

To further investigate the mechanisms driving altered biomechanical properties of healing $ScxLin^{DTR}$ tendons, bulk RNAseq analysis was conducted on days 14 and 28 post-repair samples from $ScxLin^{DTR}$ and WT. A total of three biological replicates per genotype per timepoint were submitted for analysis. At 14 days post-repair, 47 genes were up-regulated, and 313 genes were down-regulated in $ScxLin^{DTR}$ relative to WT (*Figure 6A,B*). At 28 days post-repair, 1237 genes were up-regulated, and 1296 genes were down-regulated in $ScxLin^{DTR}$ relative to WT (*Figure 6C,D*). Based on both the low number of differentially expressed genes (DEGs) and the lack of mechanical phenotype at D14 between WT and $ScxLin^{DTR}$ (*Figure 2*), our primary RNAseq analyses were focused on day 28.

## $Scx^{Lin}$ depletion drives differential expression of matrix-related gene expression

The day 28 dataset was analyzed using ingenuity pathway analysis (IPA) software. We hypothesized that a change in the matrix composition could be driving the altered biomechanical properties detected at day 28 post-repair in $ScxLin^{DTR}$ animals, possibly driven by the elevated myofibroblast content. To examine the biological effects of $Scx^{Lin}$ cell depletion at the transcriptional level, downstream effects analysis was performed by utilizing the core analysis in IPA where activation states were assigned for biological processes with p-value<0.05 and z-score $\geq$ 2 (*Table 1*). Included in the

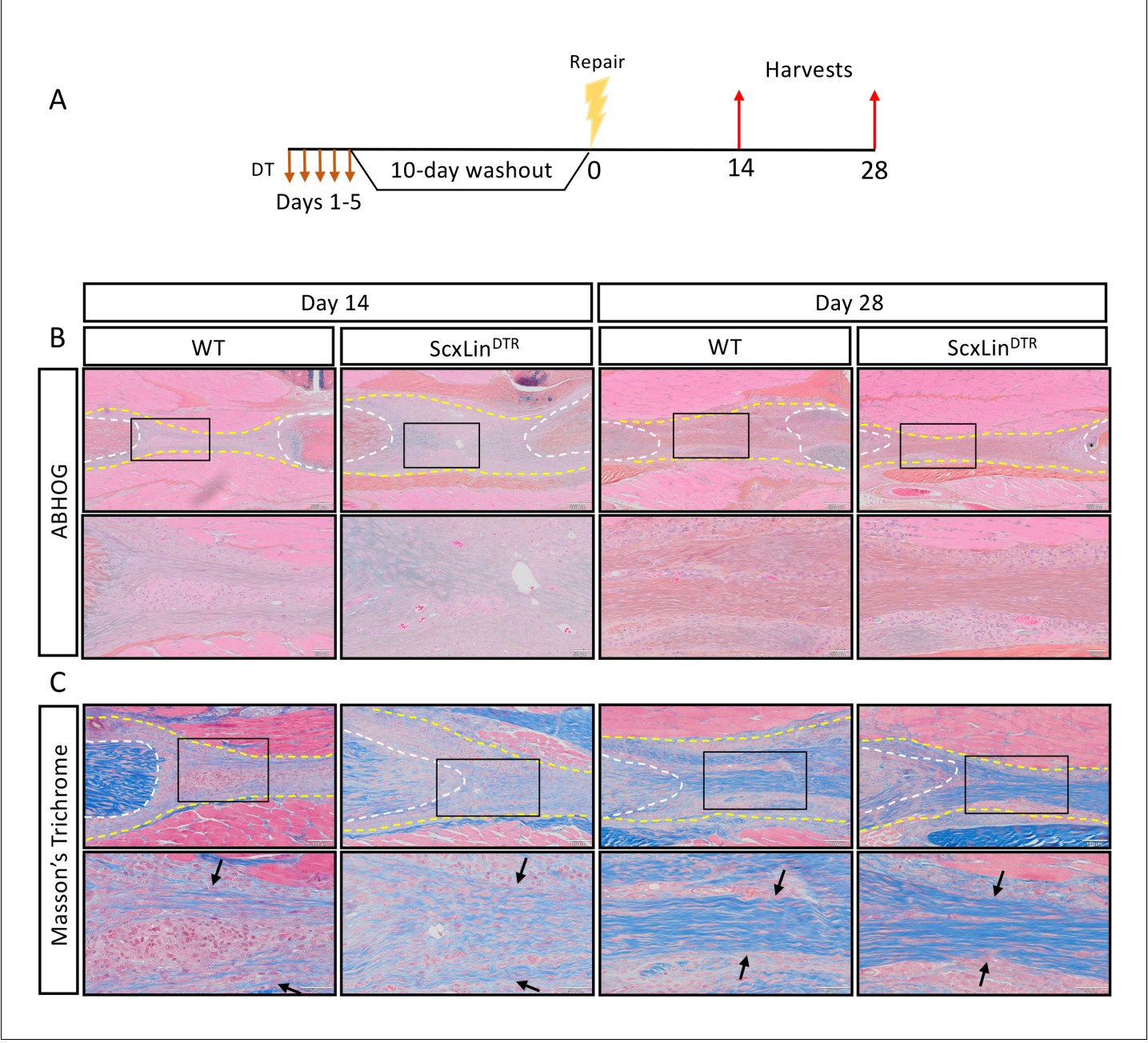

**Figure 3.** Scx[Lin] cell depletion does not disrupt formation of a bridging collagen matrix. Mice received five hindpaw injections of DT on consecutive days, underwent flexor tendon repair surgery 10 days after the final DT injection, and were harvested at 14 and 28 days post-repair (**A**). Alcian blue/ hematoxylin and Orange G stain utilized to assess overall morphology (**B**). Masson's trichrome stain used to visualize collagen content and organization (**C**). Tendon is outlined by white dotted line and scar tissue by yellow dotted line. Black boxes indicate location of higher magnification images. Boundaries of bridging collagen indicated by black arrows. N = 4 genotype per timepoint. Suture indicated by *.

significantly increased disease and function annotations was 'Fibrosis' (p=3.37E-07, Z = 2.397, *Table 1*). As we have already demonstrated that depletion of Scx[Lin] cells is not driving increased fibrotic healing (*Figure 2B,C*), we then examined if the significantly increased 'Fibrotic' annotation was indicative of altered expression of specific matrix-associated genes. Utilizing the comprehensive review of the matrisome by *Hynes and Naba, 2012*, genes coding for collagens, proteoglycans, basement membrane proteins, and glycoproteins were compiled and examined (*Table 2*). Many matrix-related genes were significantly increased in Scx[Lin]DTR repairs relative to WT littermates at

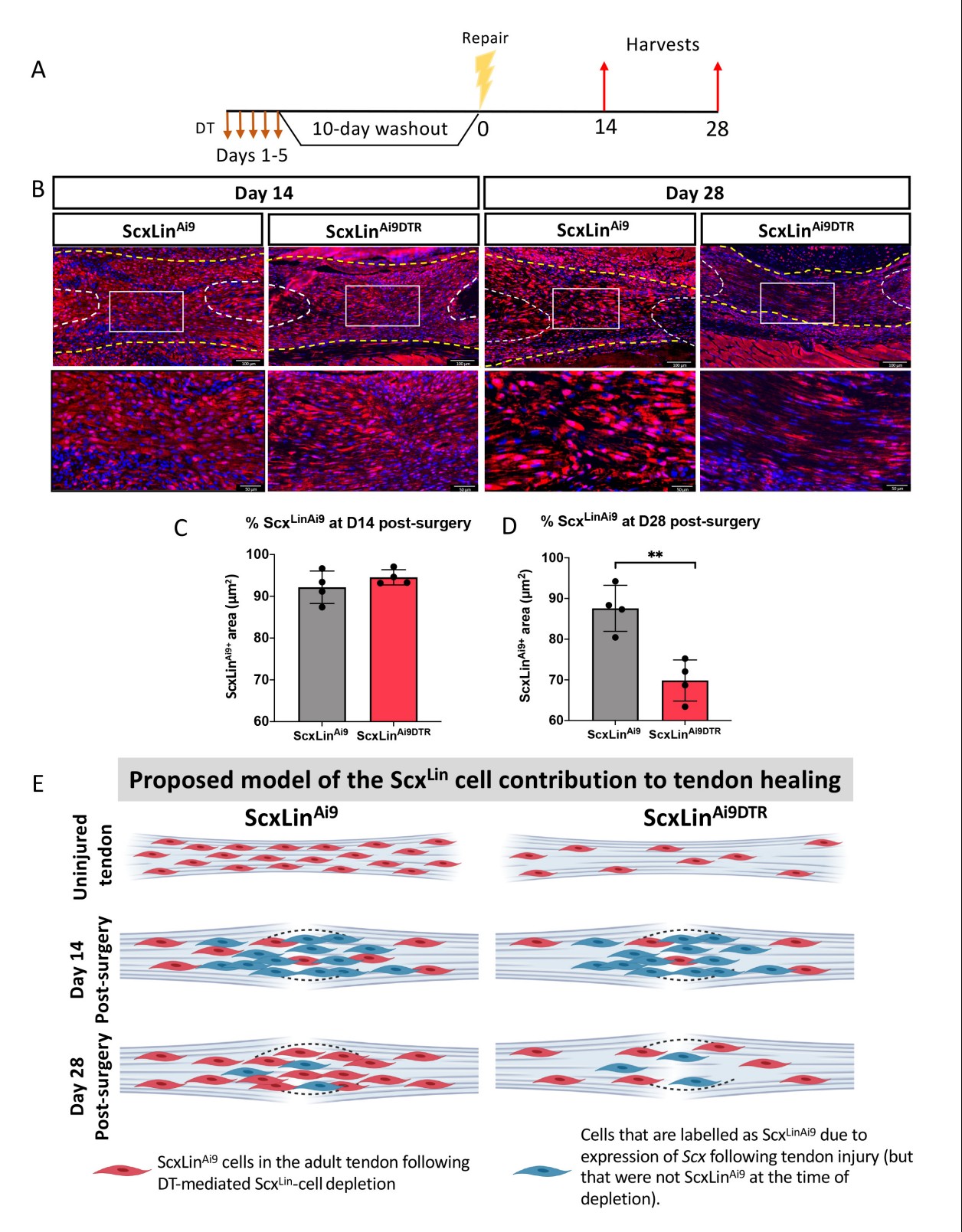

**Figure 4.** Scx[Lin] cell depletion results in time-dependent changes in ScxLin[Ai9] cell presence during tendon healing. (**A**) Mice received hind paw injections of DT on 5 consecutive days, underwent flexor tendon repair surgery 10 days after the final DT injection, and were harvested at 14 and 28 days post-repair. (**B**) Immunofluorescence for RFP (Ai9) in WT ScxLin[Ai9] and ScxLin[Ai9DTR] tendon repairs at 14 and 28 days post-repair to define changes in ScxLin[Ai9] contribution following Scx[Lin] cell depletion. Quantification of Scx[LinAi9+] area in ScxLin[Ai9] WT repairs and ScxLin[Ai9DTR] repairs at (**C**) D14 and

*Figure 4 continued on next page*

Figure 4 continued

(D) D28 post-surgery. Nuclei were stained with DAPI. N = 4 per genotype. Student's t-test used to assess statistical significance between genotypes at a given timepoint. **indicates p<0.01. (E) Proposed model of the time-dependent contributions of ScxLin[Ai9] cells to the tendon healing process. During adult tendon homeostasis ScxLin[Ai9] cells are the predominant tenocyte population and ScxLin[Ai9DTR] results in depletion of ~60% of these cells. Red cells indicate ScxLin[Ai9] cells that were present in the tendon when depletion was initiated. We hypothesize that no differences in the proportion of ScxLin[Ai9] cells is observed at D14 (concomitant with a lack of functional phenotypic differences) due to the predominance and functions of other cell populations, including those that express *Scx* in response to injury and are therefore labeled as Scx[Lin] (blue cells). In contrast, we hypothesize that by D28 the contribution of 'new' Scx[Lin] cells (blue cells) has waned, and that the ScxLin[Ai9] cells that were present in the tendon during adult tendon homeostasis (red cells) are now the predominant tenocyte population and exert their functions at this time as suggested by functional differences between WT and ScxLin[DTR] at this time. This schematic was made using http://www.biorender.com.

day 28 post-repair (ex. *Col1a2, Col3a1, Col8a1, Epyc, Thbs4*, complete list in *Table 2*), while other matrix components were significantly decreased (ex. *Col6a4, Col9a1, Fras1*, complete list in *Table 2*). To both validate the RNAseq data and to define the spatial localization of different matrix components, we performed immunofluorescence for Decorin (Dcn), Thbs4 and Mfap5. Both the staining intensity and staining extent of Dcn, Thbs4 and Mfap5 were substantially increased in D28 ScxLin[DTR] repairs, relative to WT (*Figure 6—figure supplement 1*), consistent with the increases in these matrix components identified by RNAseq (*Table 2*).

## Effects on cell motility, cytoskeletal organization, metabolism, and oxidative stress identified in ScxLin[DTR] repair tendons

Utilizing the downstream effects analysis described above (*Table 1*), we next examined other significantly altered biological functions in ScxLin[DTR] tendon repairs relative to WT. The inhibited pathways indicated decreased contractility and function of muscle (ex. 'Contractility of skeletal muscle,' p=4.96E-16, Z = −3.595) and decreased oxygen consumption ('Consumption of Oxygen,' p=9.42E-07, Z = −2.237) in ScxLin[DTR] tendons relative to wildtype controls at day 28 post-repair (*Table 1*). While only 14 disease and function annotations were found to be inhibited, 62 annotations were found to be significantly activated. Interestingly, in addition to 'Fibrosis,' the activated biological pathways indicated increased cell movement and migration (ex. 'Migration of cells,' p=6.44E-17, Z = 4.733), reorganization of the cytoskeletal network (ex. 'Organization of cytoskeleton,' p=2.94E-20, Z = 3.384), metabolism disorders (ex. 'Glucose metabolism disorder,' p=5.46E-10, Z = 3.516), and production of reactive oxygen species (ex. 'Production of reactive oxygen species,' p=6.22E-12, Z = 2.625) (*Table 1*).

## Differentially enriched canonical pathways following Scx[Lin] cell depletion prior to flexor tendon repair

To better understand signaling cascades that could be driving alterations in ScxLin[DTR] healing, enriched canonical pathways were identified using IPA core analysis. Nineteen canonical pathways were identified, where 13 were positively enriched (activated) and 6 were negatively enriched (inhibited/suppressed) (*Figure 7*, *Table 3*). Consistent with the metabolism disorders identified from the downstream effects analysis (*Table 2*), canonical pathways related to metabolism were negatively enriched (ex. 'Oxidative Phosphorylation,' -log(p)=7.6, Z = −5.303; additional pathways provided in *Figure 7* and *Table 3*). Additionally, calcium signaling was found to be negatively enriched ('Calcium Signaling,' -log(p)=5.72, Z = −2.335) (*Figure 7* and *Table 3*).

Consistent with reorganization of the cytoskeletal network identified from the downstream effects analysis (*Table 2*), numerous canonical pathways related to actin nucleation and polymerization were positively enriched (*Figure 7* and *Table 3*). These include 'Rac Signaling' (-log(p)=3.85, Z = 3.674), 'RhoA Signaling' (-log (p) = 3.97, Z = 2.502), and 'Ephrin B Signaling' (-log(p)=3.09, Z = 2.53). Other pathways that are significantly enriched in ScxLin[DTR] day 28 post-repair tendons include, but are not limited to, 'sphingosine-1-phosphate signaling' (-log(p)=3.94, Z = 2.711) and 'production of nitric oxide and reactive oxygen species in macrophages' (-log(p)=2.01, Z = 2.117) (*Figure 7* and *Table 3*).

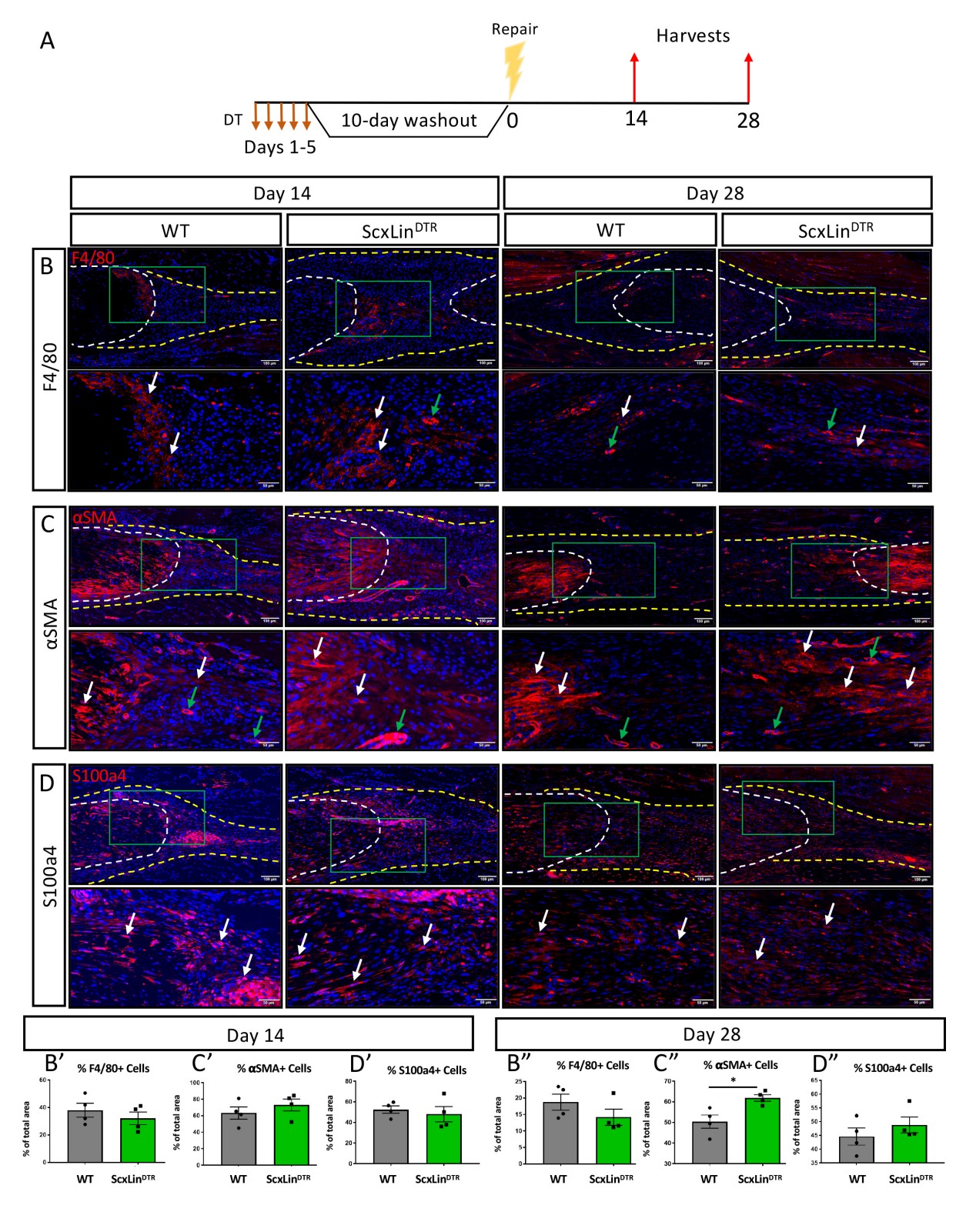

**Figure 5.** ScxLin^DTR repaired tendons heal with increased presence of αSMA+ myofibroblasts. Mice received five hindpaw injections of DT on consecutive days, underwent flexor tendon repair surgery 10 days after the final DT injection, and were harvested at 14 and 28 days post-repair (A). Immunofluorescence of WT and ScxLin^DTR repair tendons 14 and 28 days post-repair to assess F4/80+ macrophages (B), αSMA+ myofibroblasts (C), and S100a4+ cells (D). Tendon is outlined by white dotted line and scar tissue by yellow dotted line. Green boxes indicate location of higher

*Figure 5 continued on next page*

*Figure 5 continued*

magnification images. Examples of positive stain indicated by white arrows, while examples of auto-fluorescent blood cells and α-SMA+ blood vessels indicated by green arrows. Quantification of F4/80 (A' and A''), αSMA (B' and B''), and S100a4 (C' and C'') fluorescence. N = 4 per genotype per timepoint. Student's t-test used to assess statistical significance between genotypes at a given timepoint, except for D28 F4/80 and S100a4 which required a Mann-Whitney test. * indicates p<0.05.

The online version of this article includes the following figure supplement(s) for figure 5:

**Figure supplement 1.** Specific localization of αSMA staining at the tendon repair site.

## Identification of possible upstream regulators driving altered ScxLin^DTR flexor tendon healing

To identify key molecules that may be driving ScxLin^DTR tendon healing at day 28 post-repair, predicted upstream regulators were identified using IPA core analysis. Eight possible activated upstream regulators and four inhibited upstream regulators were identified (*Table 4*). The eight activated regulators included the calcium-binding protein S100a4, peptidase F2, receptors BTNL2 and F2R, transcription factors EBF2 and SOX2, the kinase NTRK2, and growth factor FGF2. The four inhibited regulators included enzyme LDHB and transcription factors FOXO4, MEF2C, and SMYD1.

## Ablation of Scx^Lin tendon cells does not significantly affect tendon post-natal growth 3 months post-ablation

In addition to roles in tendon healing, *Scx* expression is required for appropriate tendon development and growth processes (*Murchison et al., 2007*; *Gumucio et al., 2020*). However, the role of Scx+ cells during post-natal growth and adult homeostasis have not been evaluated. Local injection of DT into pre-pubescent mice (3–4 weeks old) resulted in 55% depletion of tendon cells in uninjured ScxLin^DTR FDL tendons relative to WT littermates (p=0.0007) (*Figure 8A–C*). To assess the requirement for tendon cells in post-natal growth, pre-pubescent mice who were still undergoing periods of rapid growth were injected with DT and harvested at the 3 month timepoint (ScxLin^DTR,3weeks) (*Figure 8D*). ScxLin^DTR,3weeks tendons did not exhibit an influx of F4/80+ macrophages or tendon cell differentiation into αSMA+ myofibroblasts in either genotype (*Figure 8E*); however, ScxLin^DTR mice had a 42.5% decreased tendon cell number relative to WT littermates (p=0.0454) demonstrating that the tendon cell environment was not repopulated following depletion (*Figure 8E and F*). There were no significant changes in MTP range of motion, gliding resistance, stiffness, or maximum load at failure between groups (*Figure 8G–J*). Second harmonic generation revealed no significant differences in collagen fibril dispersion between genotypes (*Figure 8K and L*). Taken together, these data suggest that Scx^Lin cells are not required for early post-natal tendon growth.

## Ablation of Scx^Lin tendon cells significantly affected tendon homeostasis 3 months post-depletion

To assess the requirement for Scx^Lin tendon cells in maintaining adult tendon homeostasis, 10–12 week-old WT and ScxLin^DTR mice were injected with DT to induce cell death and harvested after 3 months (ScxLin^DTR,10weeks) (*Figure 9A*). ScxLin^DTR,10weeks mice had a 62.3% decrease in tendon cell number relative to WT littermates (p<0.0001)(*Figure 9B and C*), demonstrating that tendon cell number had not rebounded in the three months since initial depletion (*Figure 1*). Interestingly, we consistently observed a significant accumulation of unidentified cells on the top and bottom regions of ScxLin^DTR,10weeks tendons (*Figure 9B*). Quantification of the cellular density revealed a significant increase on the top (p<0.0001) and bottom (p<0.01) regions of the ScxLin^DTR,10 weeks compared to WT littermates (*Figure 9D*). ScxLin^DTR,10weeks tendons did not exhibit an influx of F4/80+ macrophages or tendon cell differentiation into αSMA+ myofibroblasts in either genotype (*Figure 9E*). Functionally, there were no significant changes in tendon gliding function or biomechanical properties between genotypes after 3 months (*Figure 9F–I*). However, second harmonic generation imaging revealed a significant increase in overall collagen fibril dispersion in ScxLin^DTR,10weeks relative to WT littermates (WT: 8.327 degrees ± 0.39, ScxLin^DTR,10weeks: 9.815 degrees ± 0.53, p=0.0393) (*Figure 9J and K*). Based on the spatial changes in cellularity (*Figure 9D*), we also quantified fibril dispersion in a more spatially specific manner (top, middle and bottom thirds of the tendon) (*Figure 9—figure supplement 1A–D*). Interestingly, the top third of ScxLin^DTR,10weeks tendons showed a

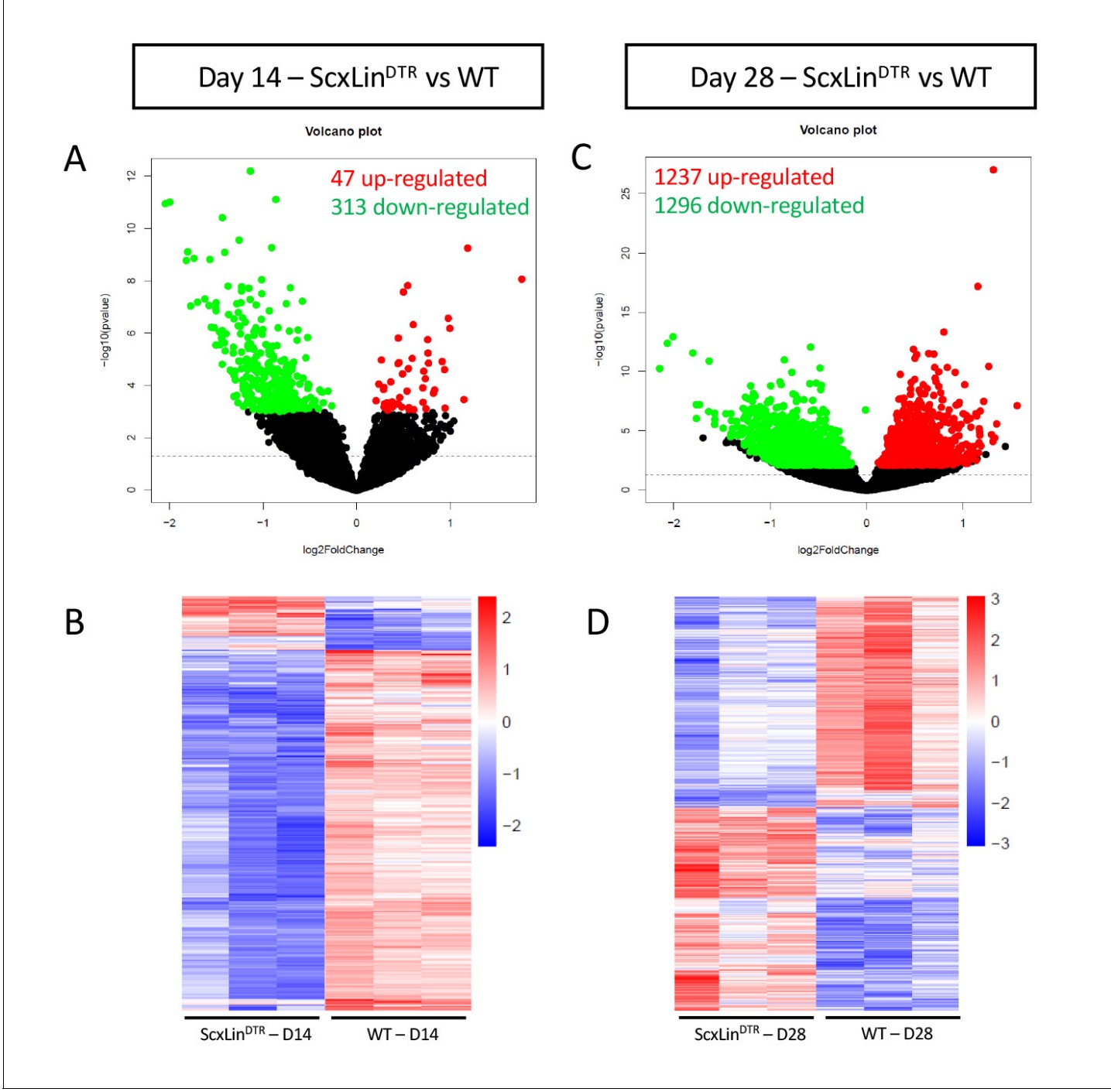

**Figure 6.** Bulk RNA sequencing reveals differences between ScxLin[DTR] and wild-type healing flexor tendons at 14 and 28 days post-repair. Representation of differentially expressed genes (DEGs) at 14 (**A, B**) and 28 (**C, D**) days post-repair. Volcano plots (**A, C**) depict significantly upregulated DEGs as red dots and significantly downregulated DEGs as green dots. DEGs are consider significant when the multiple test corrected (adjusted) p-value is $\leq$ 0.05. The dotted line represents the unadjusted p-value of 0.05. Heat maps (**B, D**) depict all significant DEGs, with the data representing the regularized log transformation of the normalized count data.

The online version of this article includes the following figure supplement(s) for figure 6:

**Figure supplement 1.** Enhanced expression specific matrix components is observed in ScxLin[DTR] tendon repairs at D28.

significant increase in dispersion compared to all regions of WT (*Figure 9—figure supplement 1E*).

**Table 1.** Ingenuity pathway analysis downstream effects - Disease and Functions.

Table of all disease and function annotations marked as significant (p≤0.05 and ABS(Z-score)>2) using IPA core analysis for ScxLin[DTR] vs WT at day 28 post-repair.

| Disease or function annotation | | p-value | Activation state | Z-Score |
|---|---|---|---|---|
| **Down-Regulated** | **Contractility of skeletal muscle** | **4.96E-16** | **Decreased** | **−3.595** |
| | Abnormal bone density | 0.000000572 | Decreased | −3.299 |
| | Contractility of muscle | 8.91E-16 | Decreased | −2.636 |
| | Intestinal cancer | 6.76E-46 | Decreased | −2.561 |
| | Bleeding | 0.00000075 | Decreased | −2.424 |
| | Malignant neoplasm of large intestine | 8.42E-46 | Decreased | −2.343 |
| | Colorectal cancer | 6.67E-23 | Decreased | −2.343 |
| | Large intestine neoplasm | 2.97E-46 | Decreased | −2.256 |
| | Colorectal tumor | 1.28E-23 | Decreased | −2.256 |
| | Function of muscle | 6.93E-14 | Decreased | −2.245 |
| | Consumption of oxygen | 0.000000942 | Decreased | −2.237 |
| | Function of skeletal muscle | 1.59E-08 | Decreased | −2.186 |
| | Intestinal tumor | 9.72E-47 | Decreased | −2.144 |
| | Development of lung carcinoma | 0.000000663 | Decreased | −2.012 |
| Up-regulated | Cell movement | 4.98E-22 | Increased | 4.735 |
| | Migration of cells | 6.44E-17 | Increased | 4.733 |
| | Cell movement of tumor cell lines | 8.66E-10 | Increased | 4.343 |
| | Reorganization of cytoskeleton | 0.000000218 | Increased | 4.296 |
| | Migration of tumor cell lines | 0.00000032 | Increased | 4.162 |
| | Engulfment of cells | 1.05E-09 | Increased | 4.057 |
| | Endocytosis | 1.2E-11 | Increased | 3.821 |
| | Leukocyte migration | 4.37E-09 | Increased | 3.821 |
| | Cell movement of blood cells | 4.04E-09 | Increased | 3.818 |
| | Homing of cells | 0.00000055 | Increased | 3.792 |
| | Formation of cellular protrusions | 1.53E-14 | Increased | 3.669 |
| | Glucose metabolism disorder | 5.46E-10 | Increased | 3.516 |
| | Invasion of cells | 9.21E-08 | Increased | 3.396 |
| | Organization of cytoplasm | 3.25E-26 | Increased | 3.384 |
| | Organization of cytoskeleton | 2.94E-20 | Increased | 3.384 |
| | Cell movement of leukocytes | 7.74E-08 | Increased | 3.38 |
| | Endocytosis by eukaryotic cells | 1.16E-08 | Increased | 3.371 |
| | Engulfment of tumor cell lines | 0.00000031 | Increased | 3.348 |
| | Proliferation of neuronal cells | 3.67E-09 | Increased | 3.313 |
| | Metabolism of carbohydrate | 1.94E-13 | Increased | 3.285 |
| | Formation of lamellipodia | 0.000000932 | Increased | 3.121 |
| | Cell movement of breast cancer cell lines | 0.000000603 | Increased | 3.103 |
| | Cell movement of fibroblast cell lines | 3.42E-08 | Increased | 3.083 |
| | Growth of neurites | 6.25E-09 | Increased | 2.981 |
| | Microtubule dynamics | 5.21E-18 | Increased | 2.974 |
| | Cell spreading | 3.31E-11 | Increased | 2.875 |
| | Formation of filopodia | 0.000000177 | Increased | 2.873 |
| | Cell movement of connective tissue cells | 0.000000415 | Increased | 2.792 |

*Table 1 continued on next page*

*Table 1 continued*

| Disease or function annotation | | p-value | Activation state | Z-Score |
|---|---|---|---|---|
| **Down-Regulated** | **Contractility of skeletal muscle** | **4.96E-16** | **Decreased** | **−3.595** |
| | Concentration of lipid | 3.14E-08 | Increased | 2.779 |
| | Progressive neurological disorder | 6.74E-10 | Increased | 2.671 |
| | Outgrowth of neurites | 7.31E-08 | Increased | 2.662 |
| | Production of reactive oxygen species | 6.22E-12 | Increased | 2.625 |
| | Quantity of macropinosomes | 0.000000865 | Increased | 2.621 |
| | Neuromuscular disease | 1.47E-15 | Increased | 2.619 |
| | Progressive myopathy | 1.28E-11 | Increased | 2.611 |
| | Synthesis of carbohydrate | 0.000000499 | Increased | 2.553 |
| | Outgrowth of cells | 0.000000025 | Increased | 2.52 |
| | Advanced malignant tumor | 0.000000208 | Increased | 2.517 |
| | Differentiation of connective tissue cells | 1.48E-09 | Increased | 2.512 |
| | Secondary tumor | 0.000000643 | Increased | 2.461 |
| | Arrhythmia | 2.14E-08 | Increased | 2.4 |
| | Fibrosis | 0.000000337 | Increased | 2.397 |
| | Extension of cellular protrusions | 0.00000098 | Increased | 2.371 |
| | Invasive tumor | 2.47E-08 | Increased | 2.345 |
| | Synthesis of reactive oxygen species | 3.6E-14 | Increased | 2.312 |
| | Organization of actin cytoskeleton | 4.37E-08 | Increased | 2.298 |
| | Disassembly of filaments | 0.000000544 | Increased | 2.27 |
| | Metabolism of reactive oxygen species | 2.61E-15 | Increased | 2.269 |
| | Cancer of cells | 9.31E-14 | Increased | 2.254 |
| | Response of tumor cell lines | 2.31E-08 | Increased | 2.231 |
| | Morphogenesis of neurons | 1.44E-12 | Increased | 2.224 |
| | Neuritogenesis | 1.86E-12 | Increased | 2.224 |
| | Neoplasia of cells | 1.76E-16 | Increased | 2.221 |
| | Quantity of metal | 0.0000002 | Increased | 2.198 |
| | Ruffling | 0.000000297 | Increased | 2.157 |
| | Tubulation of cells | 0.00000126 | Increased | 2.132 |
| | Angiogenesis | 4.02E-15 | Increased | 2.109 |
| | Hereditary myopathy | 1.58E-23 | Increased | 2.104 |
| | Dystrophy of muscle | 1.85E-11 | Increased | 2.104 |
| | Development of vasculature | 2.67E-16 | Increased | 2.06 |
| | Growth of axons | 0.000000724 | Increased | 2.017 |
| | Migration of fibroblast cell lines | 0.000000461 | Increased | 2.002 |

These data suggest a potential relationship between the increased cellularity and fibril dispersion levels on the top region of the ScxLin$^{DTR}$ FTs.

Given these changes in collagen alignment, we further assessed the ECM structure using TEM. (***Figure 9—figure supplement 2A and B***). Collagen fibril diameter distribution was substantially altered between WT (median = 147.95, Q1 = 118.73, Q3 = 179.45) and ScxLin$^{DTR,10weeks}$ (median = 213.41, Q1 = 167.49, Q3 = 261.91) (***Figure 9—figure supplement 2C***). Collagen fibril diameter of ScxLin$^{DTR,10weeks}$ increased by 30.67% compared to WT (p<0.0001) (***Figure 9—figure supplement 2D***). The collagen fibril density of ScxLin$^{DTR,10weeks}$ decreased by 48.72% compared to

**Table 2.** Regulation of matrix components in ScxLin[DTR] healing tendons at day 28.

Expression level, fold change, and adjusted p-value of key matrix-related genes in ScxLin[DTR] tendons vs WT at day 28 post-repair generated from bulk RNA-seq. Orange color indicative of increased expression and blue color indicative of decreased expression.

| Gene | BaseMean | Fold change (Log$_2$) | p-adj |
|------|----------|----------------------|-------|
| Collagens | | | |
| Col1a1 | 742982.883 | 0.275 | 0.117206 |
| Col1a2 | 720257.233 | 0.359 | 0.033355 |
| Col2a1 | 461.965 | 0.176 | 0.700754 |
| Col3a1 | 748380.28 | 0.645 | 0.005567 |
| Col4a1 | 35181.773 | 0.17 | 0.559295 |
| Col4a2 | 31627.065 | 0.265 | 0.206116 |
| Col4a3 | 49.144 | −0.009 | 0.988635 |
| Col4a4 | 143.52 | 0.232 | 0.589161 |
| Col4a5 | 284.059 | 0.372 | 0.440755 |
| Col4a6 | 52.693 | −0.093 | 0.904823 |
| Col5a1 | 118821.857 | 0.495 | 0.031814 |
| Col5a2 | 105087.042 | 0.46 | 0.092164 |
| Col5a3 | 35303.666 | 0.418 | 0.06355 |
| Col6a1 | 122410.423 | 0.406 | 0.00366 |
| Col6a2 | 122396.322 | 0.455 | 0.003183 |
| Col6a3 | 55660.629 | 0.545 | 0.021485 |
| Col6a4 | 33.599 | −0.879 | 0.0385 |
| Col6a5 | 72.046 | 0.599 | 0.286724 |
| Col6a6 | 51.234 | −0.07 | 0.901424 |
| Col7a1 | 2215.503 | −0.532 | 0.06829 |
| Col8a1 | 6553.562 | 0.983 | 0.025451 |
| Col8a2 | 4787.689 | −0.128 | 0.727708 |
| Col9a1 | 390.971 | −1.328 | 0.000857 |
| Col9a2 | 50.18 | −0.071 | 0.911757 |
| Col9a3 | 52.911 | 0.177 | 0.713852 |
| Col10a1 | 4.585 | 0.254 | N/A* |
| Col11a1 | 13080.117 | −0.239 | 0.496276 |
| Col11a2 | 1974.589 | −0.671 | 0.028981 |
| Col12a1 | 34791.768 | −0.204 | 0.545342 |
| Col13a1 | 308.833 | −0.085 | 0.853468 |
| Col14a1 | 7279.653 | 0.693 | 0.00124 |
| Col15a1 | 8889.517 | 0.385 | 0.296359 |
| Col16a1 | 24279.347 | 0.237 | 0.252556 |
| Col17a1 | 269.42 | 0.002 | N/A* |
| Col18a1 | 14754.886 | 0.391 | 0.169961 |
| Col19a1 | 4.738 | 0.133 | N/A* |
| Col20a1 | 438.526 | −0.7 | 0.060808 |
| Col22a1 | 2064 | −0.797 | 0.022217 |
| Col23a1 | 4457.185 | −0.636 | 0.025351 |
| Col24a1 | 966.961 | −0.35 | 0.047102 |
| Col25a1 | 118.033 | 0.022 | 0.975523 |

*Table 2 continued on next page*

*Table 2 continued*

| Gene | BaseMean | Fold change (Log$_2$) | p-adj |
| --- | --- | --- | --- |
| *Col26a1* | 78.9 | 0.371 | 0.475453 |
| *Col27a1* | 4062.784 | 0.406 | 0.253217 |
| *Col28a1* | 579.441 | 0.291 | 0.653743 |
| ECM proteoglycans | | | |
| *Hspg2* | 47213.867 | 0.248 | 0.108795 |
| *Aspn* | 19143.191 | 0.839 | 0.007556 |
| *Bgn* | 151131.153 | 0.251 | 0.256881 |
| *Dcn* | 95817.718 | 0.654 | 1.26E-05 |
| *Fmod* | 132295.748 | 0.172 | 0.683103 |
| *Kera* | 5207.231 | 0.522 | 0.154489 |
| *Lum* | 43114.099 | 0.353 | 0.130133 |
| *Omd* | 65.324 | 0.56 | 0.151367 |
| *Prelp* | 18853.124 | 0.381 | 0.057946 |
| *Epyc* | 133.935 | 1.187 | 0.003793 |
| *Ogn* | 5228.656 | 0.636 | 0.004079 |
| *Optc* | 30.193 | 0.042 | 0.957221 |
| *Chad* | 2558.426 | −0.133 | 0.813626 |
| *Chadl* | 146.381 | 0.377 | 0.295306 |
| *Nyx* | 21.048 | 0.443 | 0.360179 |
| *Podn* | 1763.369 | 0.432 | 0.131578 |
| *Podnl1* | 216.612 | −0.087 | 0.872241 |
| *Acan* | 8738.435 | −0.407 | 0.194754 |
| *Bcan* | 125.867 | −0.764 | 0.014902 |
| *Ncan* | 1.472 | −0.047 | N/A* |
| *Vcan* | 5214.463 | 0.431 | 0.165678 |
| *Hapln1* | 109.201 | 0.036 | 0.955062 |
| *Hapln2* | 8.84 | −0.024 | N/A* |
| *Hapln3* | 31.92 | 0.039 | 0.949531 |
| *Hapln4* | 128.699 | 0.053 | 0.926101 |
| *Prg2* | 2.717 | −0.083 | N/A* |
| *Spock1* | 12.387 | 0.674 | N/A* |
| *Spock2* | 621.489 | 0.027 | 0.956371 |
| *Spock3* | 34.16 | −0.034 | 0.963822 |
| *Prg4* | 26463.024 | −0.191 | 0.759532 |
| *Srgn* | 1500.611 | −0.08 | 0.843956 |
| *Impg2* | 49.211 | −0.196 | 0.676547 |
| *Esm1* | 120.466 | −0.02 | 0.971642 |
| Basement membrane components | | | |
| *Lama1* | 3.29 | −0.445 | N/A* |
| *Lama2* | 3430.887 | 0.361 | 0.201711 |
| *Lama3* | 104.957 | −0.303 | 0.484039 |
| *Lama4* | 7706.819 | 0.34 | 0.081572 |
| *Lama5* | 3390.382 | −0.237 | 0.430648 |
| *Lamb1* | 11272.507 | 0.236 | 0.389409 |

*Table 2 continued*

| Gene | BaseMean | Fold change (Log$_2$) | p-adj |
|---|---|---|---|
| Lamb2 | 13730.5 | 0.343 | 0.092701 |
| Lamb3 | 77.394 | −0.06 | 0.924993 |
| Lamc1 | 15292.13 | 0.313 | 0.119095 |
| Lamc2 | 424.266 | 0.393 | 0.064032 |
| Lamc3 | 21.597 | 0.093 | 0.886633 |
| Nid1 | 12717.306 | 0.539 | 0.02301 |
| Nid2 | 2799.817 | 0.179 | 0.586124 |
| Colq | 358.747 | −0.379 | 0.341156 |
| Major ECM glycoproteins | | | |
| Eln | 17607.464 | 0.518 | 0.270402 |
| Emilin1 | 6960.539 | 0.124 | 0.684821 |
| Emilin2 | 4405.463 | 0.512 | 0.061758 |
| Emilin3 | 595.356 | 0.857 | 0.005562 |
| Emid1 | 550.685 | 0.252 | 0.631487 |
| Fbln1 | 2214.023 | 0.23 | 0.320082 |
| Fbln2 | 48333.486 | 0.127 | 0.615495 |
| Fbln5 | 2725.675 | 0.326 | 0.023116 |
| Fbln7 | 4562.409 | 0.784 | 0.00233 |
| Efemp1 | 1308.939 | 0.543 | 0.077648 |
| Efemp2 | 4858.991 | 0.222 | 0.11327 |
| Fbn1 | 36959.196 | 0.668 | 0.005087 |
| Fbn2 | 2856.248 | −0.008 | 0.983466 |
| Fn1 | 510510.053 | 0.307 | 0.223595 |
| Fras1 | 430.729 | −0.837 | 0.013797 |
| Gldn | 464.82 | 0.424 | 0.390168 |
| Hmcn1 | 1034.806 | 0.589 | 0.036404 |
| Hmcn2 | 6034.922 | 0.252 | 0.456036 |
| Ibsp | 2.365 | −0.125 | N/A* |
| Matn1 | 0.671 | 0.074 | N/A* |
| Matn2 | 6213.381 | 0.503 | 0.015668 |
| Matn3 | 201.369 | −0.581 | 0.155595 |
| Matn4 | 2796.498 | −0.245 | 0.6134 |
| Mfap1a | 536.887 | 0.101 | 0.639732 |
| Mfap1b | 398.958 | 0.03 | 0.902813 |
| Mfap2 | 3608.727 | 0.465 | 0.013072 |
| Mfap3 | 1213.924 | 0.119 | 0.519005 |
| Mfap4 | 4217.93 | 0.4 | 0.210242 |
| Mfap5 | 14548.008 | 0.753 | 5.34E-05 |
| Mmrn1 | 439.183 | 0.492 | 0.262418 |
| Mmrn2 | 1557.776 | −0.201 | 0.519886 |
| Npnt | 1084.388 | −0.453 | 0.209896 |
| Papln | 16.561 | −0.122 | 0.859483 |
| Postn | 102294.871 | 0.591 | 0.048873 |
| Sparc | 331616.177 | 0.479 | 0.000296 |

*Table 2 continued on next page*

*Table 2 continued*

| Gene | BaseMean | Fold change (Log$_2$) | p-adj |
|---|---|---|---|
| *Sparcl1* | 12872.967 | −0.027 | 0.959288 |
| *Spp1* | 29368.724 | 0.234 | 0.637831 |
| *Srpx2* | 5778.794 | 0.362 | 0.077229 |
| *Tnc* | 28609.297 | 0.378 | 0.326135 |
| *Tnn* | 5098.501 | −0.623 | 0.126873 |
| *Tnr* | 19.271 | 0.482 | 0.385953 |
| *Tnxa* | 6.323 | 0.944 | N/A* |
| *Tnxb* | 18888.425 | 0.781 | 0.015263 |
| *Thbs1* | 14477.798 | 0.49 | 0.099587 |
| *Thbs2* | 36613.787 | 0.39 | 0.149809 |
| *Thbs3* | 17595.855 | 0.267 | 0.193149 |
| *Thbs4* | 203095.542 | 0.841 | 5.69E-05 |
| *Comp* | 35501.759 | 0.293 | 0.10314 |

WT (p<0.0001) (*Figure 9—figure supplement 2E*). Finally, both the WT and the ScxLin$^{DTR,10weeks}$ groups exhibited similar levels of fibril irregularity (p=0.9023) (*Figure 9—figure supplement 1F*).

## Discussion

Previous work has established the importance of *Scx* expression in tendon development (*Murchison et al., 2007*), growth (*Gumucio et al., 2020*), and healing (*Sakabe et al., 2018*), but few studies have considered the direct contributions of Scx$^{Lin}$ tendon cells to these processes. In the present study, we examined the function of Scx$^{Lin}$ tendon cells during adult flexor tendon healing and made the surprising discovery that Scx$^{Lin}$ cell depletion prior to tendon injury and repair significantly enhanced biomechanical properties by day 28 post-repair. We characterized key cell populations known to be important in healing and regeneration and utilized RNA-Seq to investigate the mechanism driving these biomechanical changes. Lastly, we examined the effects of Scx$^{Lin}$ cell depletion on post-natal tendon growth and adult tendon homeostasis and found that Scx$^{Lin}$ cells are required for maintenance of collagen ECM alignment and organization in adult, but not early post-natal flexor tendon growth.

Tendon cell depletion had surprisingly beneficial effects on healing, with biomechanical properties significantly increased at day 28 post-repair relative to WT, with no impact on gliding function. These results indicate that the improved biomechanical properties are likely not due to increased levels of disorganized matrix/scar within the healing tendon, but that the healing process may be shifted toward a regenerative phenotype. Equally striking is that the significant improvements in biomechanical properties did not emerge until 28 days post-repair, which is firmly into the remodeling phase of healing and is consistent with the low DEG number at day 14 post-repair relative to day 28. This suggests that Scx$^{Lin}$ cells are important in the late proliferative-remodeling phases of healing and possibly enact their greatest effects by modulating the remodeling process. Consistent with this is the lack of difference in proportion of ScxLin$^{Ai9}$ cells at D14 between WT and ScxLin$^{Ai9DTR}$, while a significant decrease in ScxLin$^{Ai9}$ cells was observed in ScxLin$^{Ai9DTR}$ tendon repairs, relative to WT, at D28. While further studies are needed to completely define this process, these data suggest that 'new' Scx$^{Lin}$ cells (e.g. those that express *Scx* in response to tendon injury and are therefore labeled as ScxLin$^{Ai9}$, but that were not Scx$^{LinAi9}$ at the time of depletion) may predominate during early healing, so the effects of depleting Scx$^{Lin}$ prior to injury are minimal. However, by D28 these 'new' ScxLin$^{Ai9}$ cells may undergo apoptosis and be cleared during progressive tissue remodeling, such that the cells that remain at D28 are primarily derived from the ScxLin$^{Ai9}$ cells present in the adult tendon prior to injury. Therefore, the effects of ScxLin$^{Ai9DTR}$ become more apparent and allow interrogation of the functional effects on healing of depleting adult tendon resident Scx$^{Lin}$ cells by D28. Finally, it

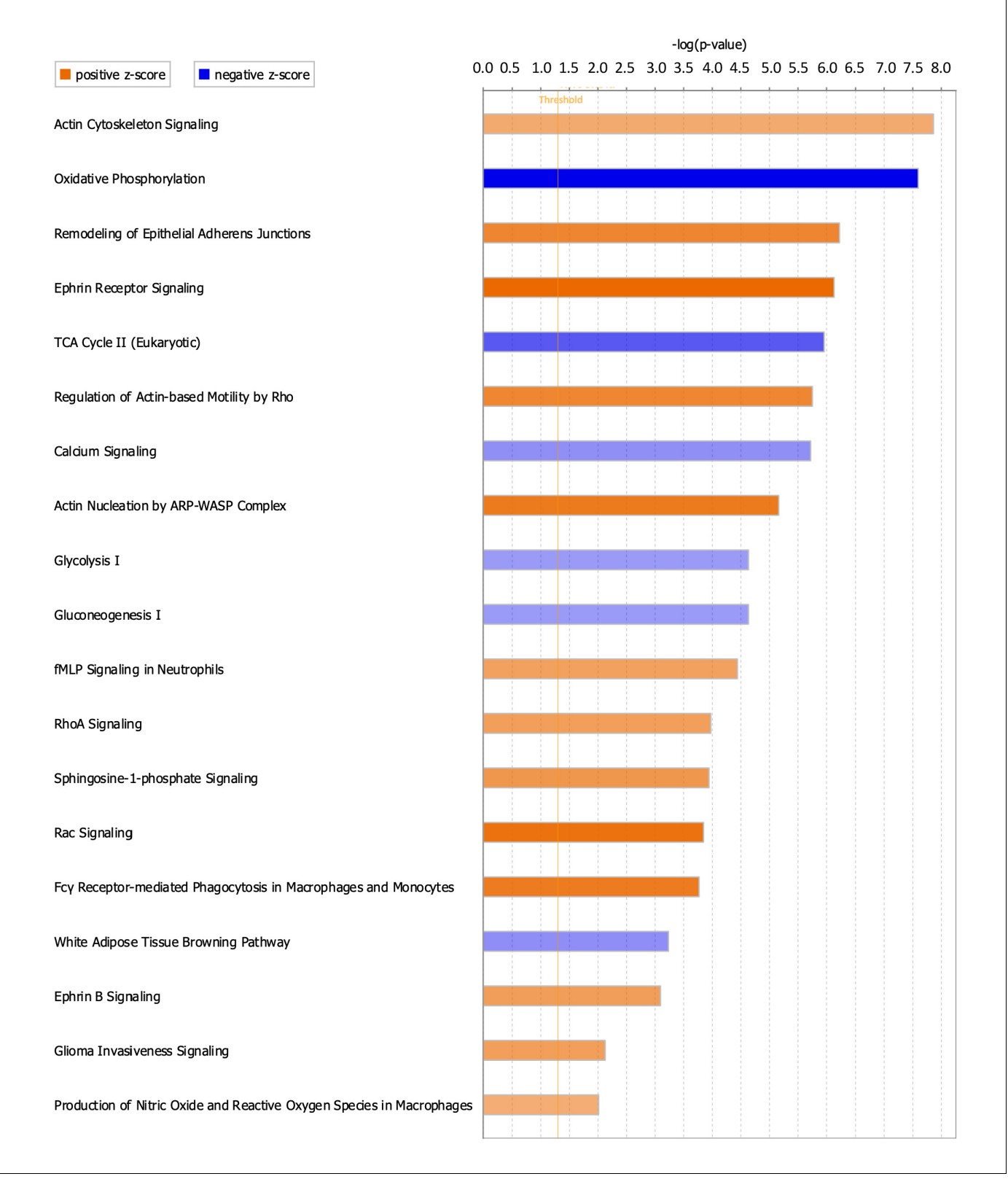

**Figure 7.** Canonical pathways positively and negatively enriched in ScxLin$^{DTR}$ healing tendons at day 28 post-repair. Ingenuity pathway analysis was utilized to determine positively and negatively significantly enriched pathways in ScxLin$^{DTR}$ healing tendons at day 28 post-repair. Canonical pathways were considered significant if p<0.05 and ABS(Z-score)>2. The orange color indicates pathways that are significantly, positively enriched ('activated'), *Figure 7 continued on next page*

*Figure 7 continued*

while the blue color indicates significantly, negatively enriched ('inhibited') pathways. The orange dotted line represents -log (1.3)=0.05, indicating the p-value cut-off.

is also possible that there is a compensatory response by other tenocyte subpopulations. While we do not see any changes in S100a4[+] cells, it is possible that there may be an increase in S100a4-lineage cells, which can express *Scx* during tendon healing (*Best and Loiselle, 2019*), or other cells.

Our RNA-Seq results revealed significant changes in matrix-related gene expression at day 28 post-repair, suggesting that the matrix composition of the healing tendon may be substantially altered in the absence of Scx[Lin] cells, with a shift toward either a more regenerative or mature matrix composition. Future proteomics analysis and additional experimentation will be required to further examine the matrix differences between ScxLin[DTR] and WT healing flexor tendons. It has previously been reported that tendon strength remains significantly decreased relative to uninjured controls 63 days post-repair using this flexor tendon repair model in C57Bl/6J mice (*Loiselle et al., 2009*). Therefore, it could be beneficial to examine time points beyond day 28 to see if ScxLin[DTR] tendons reach equivalency with uninjured controls. The improved biomechanical properties contrasted with our original hypothesis which predicted that *Scx*[Lin] cells would be necessary for proper healing and would contribute to formation of the bridging collagen tissue. Both ScxLin[DTR] and WT mice exhibited a collagen bridge at 14- and 28 days post-repair suggesting that the Scx[Lin] cells targeted for depletion prior to injury are not the predominant cell population laying down this collagenous matrix. This is surprising due to studies demonstrating that *Scx* promotes collagen formation (*Sakabe et al., 2018*; *Leéjard et al., 2007*), highlighting the important distinction between Scx[Lin] cells and active *Scx* expression.

While myofibroblast persistence is considered a primary driver of fibrosis (*Hinz and Lagares, 2020*), recent work has revealed that myofibroblasts are a highly heterogenous population with

**Table 3.** Ingenuity pathway analysis canonical pathways.

All enriched pathways marked as significant (-log(p-value)>1.3 and ABS(Z-score)>2) using IPA core analysis for ScxLin[DTR] vs WT at day 28 post-repair.

| Canonical pathway | | -log(p) | Z-Score |
|---|---|---|---|
| **Negatively enriched** | **Oxidative phosphorylation** | **7.6** | **−5.303** |
| | TCA Cycle II (Eukaryotic) | 5.96 | −3.464 |
| | White Adipose Tissue Browning Pathway | 3.23 | −2.353 |
| | Calcium Signaling | 5.72 | −2.335 |
| | Glycolysis I | 4.63 | −2.111 |
| | Gluconeogenesis I | 4.63 | −2.111 |
| Positively enriched | Ephrin Receptor Signaling | 6.13 | 3.888 |
| | Rac Signaling | 3.85 | 3.674 |
| | Actin Nucleation by ARP-WASP Complex | 5.16 | 3.441 |
| | Fcγ Receptor-mediated Phagocytosis in Macrophages and Monocytes | 3.77 | 3.411 |
| | Remodeling of Epithelial Adherens Junctions | 6.23 | 3.162 |
| | Regulation of Actin-based Motility by Rho | 5.75 | 3.128 |
| | Sphingosine-1-phosphate Signaling | 3.94 | 2.711 |
| | Ephrin B Signaling | 3.09 | 2.53 |
| | RhoA Signaling | 3.97 | 2.502 |
| | Glioma Invasiveness Signaling | 2.13 | 2.496 |
| | fMLP Signaling in Neutrophils | 4.44 | 2.449 |
| | Actin Cytoskeleton Signaling | 7.86 | 2.214 |
| | Production of Nitric Oxide and Reactive Oxygen Species in Macrophages | 2.01 | 2.117 |

**Table 4.** Ingenuity pathway analysis upstream regulators.

All possible upstream regulators where expression log ratio $\geq$ 0.5, ABS(Z-score)>2, p-value<0.05, and agreement between predicted activation state and directionality of regulator's gene expression, compiled using IPA core analysis for ScxLin$^{DTR}$ vs WT at day 28 post-repair.

| Upstream regulator | Expression log ratio | Predicted activation state | Activation Z-score | p-value of overlap |
|---|---|---|---|---|
| Activated in ScxLin$^{DTR}$ | | | | |
| S100A4 | 0.518 | Activated | 2.946 | 0.00456 |
| F2 | 0.533 | Activated | 2.606 | 0.000134 |
| BTNL2 | 0.805 | Activated | 2.324 | 0.012 |
| EBF2 | 0.657 | Activated | 2.223 | 0.000738 |
| F2R | 0.599 | Activated | 2.22 | 0.0243 |
| NTRK2 | 0.608 | Activated | 2.137 | 0.012 |
| SOX2 | 1.138 | Activated | 2.071 | 0.0000143 |
| FGF2 | 0.544 | Activated | 2.017 | 0.000393 |
| Inhibited in ScxLin$^{DTR}$ | | | | |
| FOXO4 | −0.501 | Inhibited | −2.697 | 0.0236 |
| MEF2C | −0.905 | Inhibited | −2.577 | 3.06E-08 |
| SMYD1 | −0.836 | Inhibited | −2.219 | 2.63E-12 |
| LDHB | −0.609 | Inhibited | −2.219 | 0.000241 |

differences in pro-fibrotic markers, gene expression, and cross-linking ability, suggesting that myofibroblasts contribute to both fibrotic and regenerative processes (*Shook et al., 2018*). Despite the elevated myofibroblast presence in ScxLin$^{DTR}$ repairs at day 28, these tendons healed with increased biomechanical properties while experiencing no deficits in tendon range of motion, suggesting a regenerative rather than fibrotic healing process. Future work to comprehensively define the myofibroblast landscape in both WT and ScxLin$^{DTR}$ mice is necessary to determine if ScxLin$^{DTR}$ myofibroblasts are more 'pro-regenerative' than those present in WT repairs. Similarly, understanding how myofibroblast subtypes affect matrix deposition at the repair site represents an important area of future study. Related to this, previous work has shown that regeneration is mediated in large part by the immune response, including macrophages (*Vagnozzi et al., 2020*). However, we did not see any differences in overall macrophage content, or in pathways related to immune response in the RNA-seq. Future work, including single-cell RNA sequencing studies will be important to provide a comprehensive understanding of how Scx$^{Lin}$ cell depletion alters the overall cellular environment and how this dictates the functional changes that are observed.

The RNA-seq data suggests that healing ScxLin$^{DTR}$ tendons exhibit less overall metabolic activity compared to WT repairs at day 28. This is highlighted in IPA's canonical pathways core analysis, where 'Oxidative Phosphorylation,' 'TCA Cycle II,' 'Glycolysis I,' and 'Gluconeogenesis I' were all predicted to be negatively enriched ('suppressed') (*Figure 7, Table 3*). Additionally, IPA's disease and function core analysis predicts an inhibitory effect on 'Consumption of oxygen,' while simultaneously predicting 'Metabolism of carbohydrate' activation, increased 'Synthesis of carbohydrate,' and 'Glucose metabolism disorder (*Table 1*)." It has previously been demonstrated in a variety of tissues that metabolic reprogramming is important for cellular differentiation and regeneration (*Chen et al., 2019*; *De Santa et al., 2019*; *Osuma et al., 2018*; *Lai et al., 2019*). In addition to the dysregulation of metabolism, our RNA-seq data also revealed an increase in 'Production of reactive oxygen species', 'Synthesis of reactive oxygen species', and 'Metabolism of reactive oxygen species (*Table 1*)'. Despite reactive oxygen species (ROS) being implicated in various pathologies (*Schieber and Chandel, 2014*), recent studies have also identified functional roles for ROS in regeneration (*Santabárbara-Ruiz et al., 2019*; *Youm et al., 2019*; *Labit et al., 2018*; *Santabárbara-Ruiz et al., 2015*). While little is currently understood about the role of metabolism and ROS in tendon healing, these data clearly identify how changes in the cellular composition of the healing tendon can shift both the metabolic profile and the healing program. As such, investigating metabolic

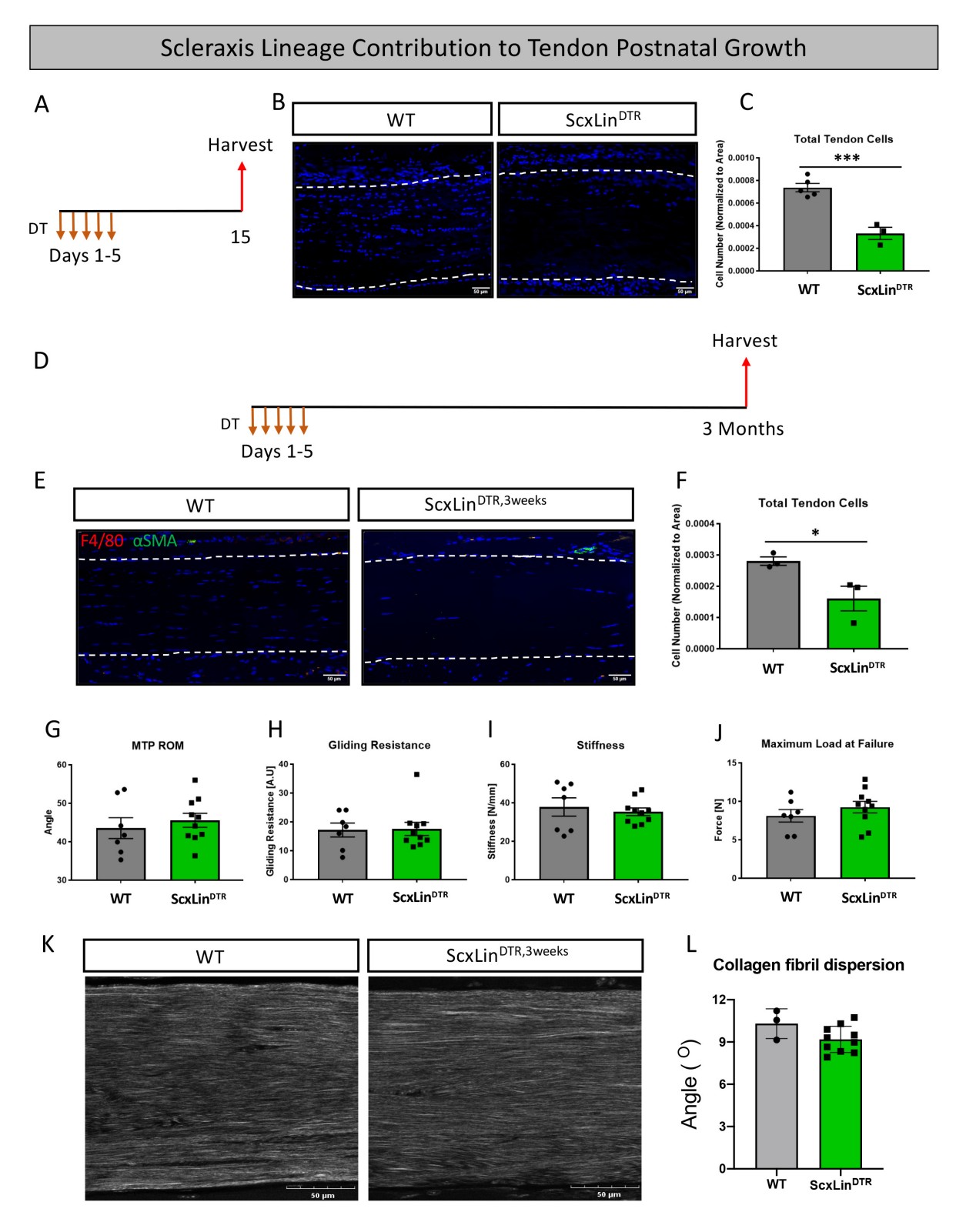

**Figure 8.** Tendon cell ablation does not negatively affect post-natal tendon growth 3 months post-ablation. Pre-pubescent mice (3–4 weeks old) received five hindpaw injections of DT and were harvested 10 days after the final injection to assess tendon cell depletion (ScxLin^DTR) (**A**). Hindpaw sections from both WT and ScxLin^DTR hindpaws (**B**). Quantification of WT and ScxLin^DTR,3weeks tendon cell number in pre-pubescent mice (**C**). To assess effects of tendon cell depletion on post-natal tendon growth, mice received five hindpaw injections of DT on consecutive days at 3–4 weeks of age and

*Figure 8 continued on next page*

*Figure 8 continued*

were harvested uninjured 3 months later for biomechanical, gliding, and histological evaluation (ScxLin[DTR,3weeks]) (D). Co-immunofluorescence of F4/80 (macrophages) and αSMA (myofibroblasts) in uninjured WT and ScxLin[DTR,3weeks] tendons (E). Quantification of WT and ScxLin[DTR,3weeks] tendon cell number (F). Measurement of metatarsophalangeal (MTP) joint flexion angle (G), gliding resistance (H), stiffness (I), and maximum load at failure (J) of WT and ScxLin[DTR,3weeks] uninjured tendons. N = 7–10 per genotype. Second harmonic generation (K) and quantification (L) of collagen fibril dispersion of WT and ScxLin[DTR,3weeks]. N = 3 per genotype. Nuclei stained with DAPI. Tendon is outlined by white dotted lines. Student's t-test used to assess statistical significance between genotypes. * indicates $p < 0.05$, *** indicates $p < 0.001$.

reprogramming and the dynamic roles of ROS in acute tendon healing represents an exciting area of future work.

In addition to investigating the role of tendon cells during healing, we also utilized ScxLin[DTR] mice to assess how tendon cell ablation affected post-natal tendon growth (ScxLin[DTR,3weeks]) and adult tendon homeostasis (ScxLin[DTR,10weeks]). Although there were no significant differences in gliding function or biomechanical properties between ScxLin[DTR] and WT genotypes in either age group, ScxLin[DTR,10weeks] animals exhibited significantly increased overall collagen fibril dispersion, as well as significant changes in dispersion in specific regions of the tendon. While it is unclear what is driving these spatially-specific changes in fibril organization, it is notable that the most profound changes occur in the top epitenon region rather than endotenon, and that this area is consistent with altered and increased cellularity in ScxLin[DTR,10weeks] tendons, though the identity and function of these cells are as yet unknown. In addition to matrix alignment, ScxLin[DTR, 10 weeks] also increased collagen fibril diameter as measured by TEM. Interestingly, one potential interpretation of the TEM data relates to the loss of proteoglycans (PGs) on the surface of the collagen fibrils, which normally prevents lateral fusion of collagen fibrils. Numerous in vitro and in vivo studies have shown the importance of the PGs to maintain the normal collagen fibril structure. Using proteinase treatment to remove surface bound PGs, Graham et al., demonstrated lateral collagen fibril aggregation, resulting in fused fibrils with increased diameter (*Graham et al., 2000*). In addition, Decorin-/- tail tendons also demonstrated lateral fusion and increased fibril diameter, relative to WT (*Danielson et al., 1997*). While there is no direct evidence of PG production by Scx+ cells, Scx-/- hearts have decreased PG content (*Barnette et al., 2013*; *Barnette et al., 2014*), while Sakabe et al., found that Scx-/- tendons lacked fibromodulin production during tendon healing (*Sakabe et al., 2018*). Moreover, recent work has shown that Scx knockdown in adult equine tendon cells alters the PG environment (*Paterson et al., 2020*). Collectively, these data suggest that the increase in fibril diameter observed in ScxLin[DTR,10-weeks] could be due to diminution of PG production following depletion of Scx[Lin] cells. However, we have not directly tested this, and future proteomic studies will be required to determine if and which specific PGs may be altered following Scx[Lin] cell depletion. Collectively, these data indicate that loss of Scx[Lin] cells may be detrimental to tendon tissue maintenance, but that negative biomechanical effects are not yet manifested 3 months post-depletion. Future studies looking beyond the 3-month time-point will be informative to understand the role of tendon cells on maintenance of tendon tissue. It is also possible that negative effects on tendon biomechanical properties could be the result of Scx[Lin] cell death in non-tendon tissues, such as muscle in the hindpaw (*Mendias et al., 2012*). Despite nearly identical depletion efficiencies between ScxLin[DTR,3weeks] and ScxLin[DTR,10weeks] animals, only ScxLin[DTR,10weeks] tendons exhibited collagen disorganization and differences in fibril size. This suggests possible differences in tendon cell sub-populations present during growth and homeostasis, as well as the potential contribution of extrinsic progenitor populations to tendon growth (*Dyment et al., 2014*), such that depletion could differentially impact post-natal growth and adult tendon homeostasis. Moreover, it is possible that ScxLin[DTR,3weeks] did not disrupt tendon growth due to more rapid compensation by non-depleted Scx[Lin] cells, or the addition of 'new' Scx[Lin] cells during this period of rapid tissue growth.

We had initially planned to deplete tendon cells using the inducible Scx-Cre[ERT2] crossed to the diphtheria toxin A mouse (*Voehringer et al., 2008*), but insufficient recombination occurred, and targeted cell death was not achieved. Therefore, to successfully deplete tendon cells, Scx-Cre mice were crossed to a diphtheria toxin receptor mouse model (ScxLin[DTR]) (*Buch et al., 2005*). The initial attempt to deplete cells using this model employed a series of intraperitoneal injections (200 ng/day) and resulted in all ScxLin[DTR] mice dying within four days of the initial injection while WT animals were unaffected. ScxLin[DTR] mice likely died due to apoptosis of non-tendon/ligament associated

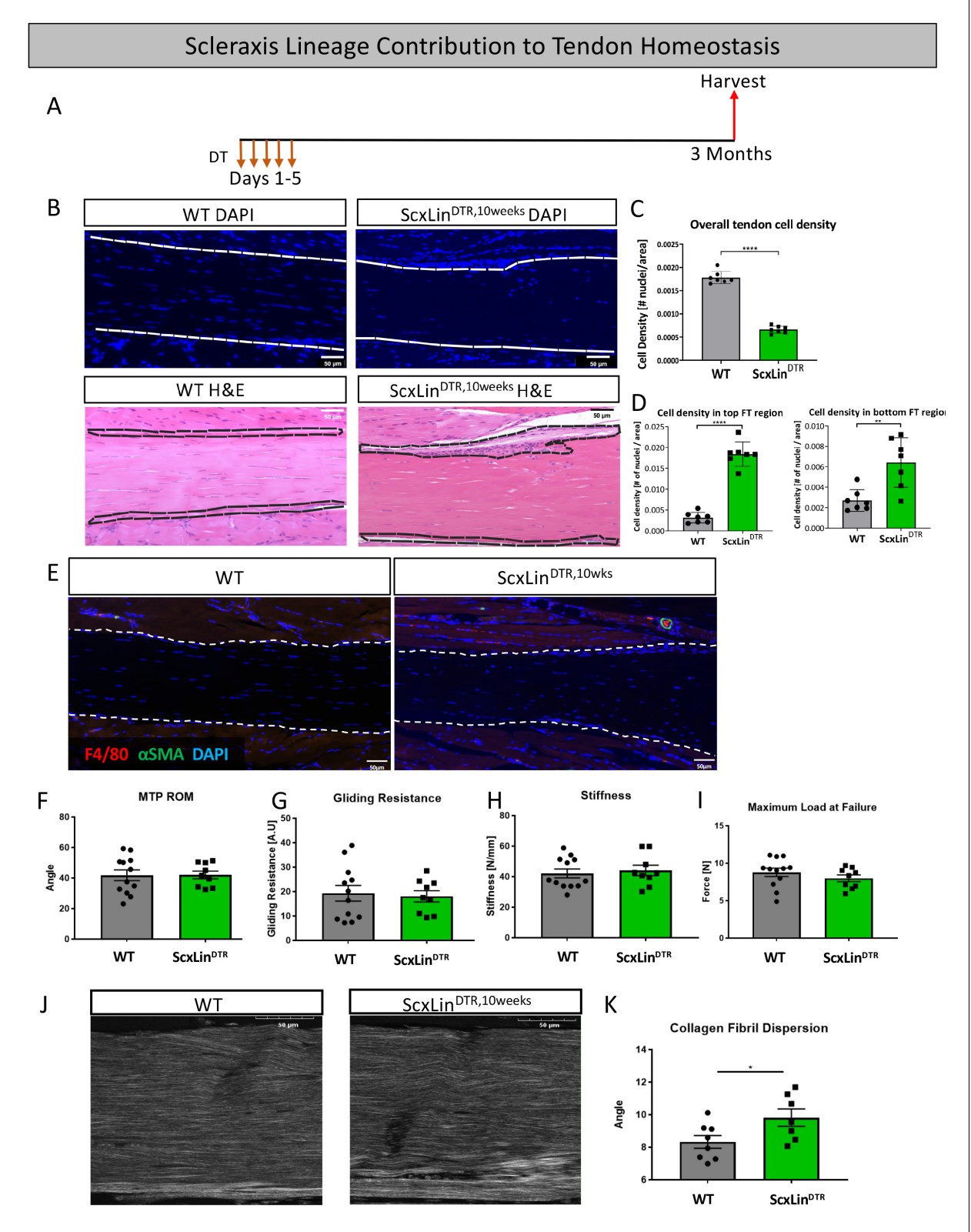

**Figure 9.** Tendon cell ablation negatively affected tendon homeostasis 3 months post-ablation. Mice received five hindpaw injections of DT on consecutive days at 10–12 weeks of age and were harvested uninjured 3 months later for biomechanical, gliding, and histological evaluation (ScxLin^DTR,10weeks) (**A**). Cellularity was assessed using DAPI (**B**) and quantified (**C**) 3 months after Scx^Lin cell depletion. H and E staining was used to better define the hypercellular regions near the tendon epitenon. Cell density was quantified at the top and bottom boundaries of the tendon (**D**). *Figure 9 continued on next page*

*Figure 9 continued*

N = 7 per genotype. Co-immunofluorescence of F4/80 (macrophages) and αSMA (myofibroblasts) in uninjured WT and ScxLin<sup>DTR</sup> tendons (E). N = 3 per genotype. Measurement of metatarsophalangeal (MTP) joint flexion angle (F), gliding resistance (G), stiffness (H), and maximum load at failure (I) of WT and ScxLin<sup>DTR,10weeks</sup> uninjured tendons. N = 9–12 per genotype. Second harmonic generation (SHG) (J) and quantification (K) of collagen fibril dispersion of WT and ScxLin<sup>DTR,10weeks</sup>. N = 7–8 per genotype. Nuclei stained with DAPI. Tendon is outlined by white dotted lines. Student's t-test used to assess statistical significance between genotypes. * indicates p<0.05.

The online version of this article includes the following figure supplement(s) for figure 9:

**Figure supplement 1.** Collagen fibrils in the top third FT region have an altered organization at 3 months post-depletion.

**Figure supplement 2.** Collagen fibrils exhibit an altered diameter and density at 3 months post-depletion.

Scx<sup>Lin</sup> cells. For example, it has previously been shown that *Scx* is expressed in the lung (*Pryce et al., 2007*; *Perez et al., 2003*), kidney (*Pryce et al., 2007*), muscle (*Mendias et al., 2012*), and brain, (*Perez et al., 2003*) among others. Therefore, to successfully utilize this model of cell depletion while simultaneously preventing ScxLin<sup>DTR</sup>-associated death, a series of low-dose (20 ng/day) local hind paw injections were administered. We successfully utilized this model of in vivo tendon cell depletion and reached a depletion level of 57%. Therefore, the ScxLin<sup>DTR</sup> data should be viewed as a model of partial depletion as ~40% of tendon cells remained using this approach. Future work utilizing increased or prolonged DT treatment could be attempted to obtain more complete tendon cell depletion, and to determine if the beneficial effects of tendon cell depletion may be reversed with complete ablation of Scx<sup>Lin</sup> cells from the tendon. Additionally, our current ScxLin<sup>DTR</sup> ablation model targets cells prior to injury and thus does not target any cells that turn on *Scx* after injury. It has previously been shown that *Scx* knockout during Achilles tendon healing drives incomplete remodeling of type III collagen to type I collagen (*Sakabe et al., 2018*). As such, it could be interesting to assess if depletion of cells actively expressing *Scx* during healing resulted in deleterious effects on healing. However, local DT injection into the hind paw following repair surgery represents a technical challenge as repeated injections to the healing tendon is likely to disrupt the normal healing process. Moreover, the overlap between those Scx<sup>Lin</sup> cells in the adult tendon prior to injury, and those that express *Scx* after injury is not yet defined. Finally, while there are likely to be some differences between tendons in terms of the functional contribution of Scx<sup>Lin</sup> cells to healing process, the type of injury model used is also likely to impact data interpretation. For example, adult Scx<sup>Lin</sup> cells do not contribute to tissue bridging in a model of Achilles tendon transection without repair (*Howell et al., 2017*), but are found in the bridging tissue following flexor tendon injury and repair (*Best and Loiselle, 2019*). As such, future studies will be needed to clarify if, or how, the functional contribution of Scx<sup>Lin</sup> cells differ between tendons and injury models.

Altogether, these data demonstrate that Scx<sup>Lin</sup> cell depletion is beneficial for tendon healing as it increases biomechanical properties several weeks after repair, possibly due to alterations in matrix composition, deposition, and/ or remodeling by αSMA+ myofibroblasts. Moreover, we have identified several molecular pathways that are altered following Scx<sup>Lin</sup> cell depletion, such as changes in metabolism, cell motility, and the cytoskeleton, which may represent important targets for successfully modulating the healing process. Finally, Scx<sup>Lin</sup> depletion does not significantly disrupt the biomechanical properties of tendons during early post-natal growth or adult tendon homeostasis within three months of Scx<sup>Lin</sup> cell depletion. However, given the changes in matrix alignment and organization in adult ScxLin<sup>DTR</sup> tendons, it is possible that mechanical changes may occur following longer periods of depletion. Understanding the complex nature of tendon cells during healing, growth, and homeostasis can provide us with insights for driving regenerative healing and healthy tendon maintenance in the future.

## Materials and methods

**Key resources table**

| Reagent type (species) or resource | Designation | Source or reference | Identifiers | Additional information |
| --- | --- | --- | --- | --- |

*Continued on next page*

*Continued*

| Reagent type (species) or resource | Designation | Source or reference | Identifiers | Additional information |
|---|---|---|---|---|
| Genetic reagent (*Mus musculus*) | *Scx-Cre* | Dr. Ronen Schweitzer | MGI:5317938 | Referred to as Scx<sup>Lin</sup> in manuscript |
| Genetic reagent (*Mus musculus*) | C57BL/6-Gt(ROSA)26 Sor<sup>tm1(HBEGF)Awai</sup>/J (*Rosa-DTR<sup>LSL</sup>*) | Jackson Laboratory | Stock #: 007900 RRID:IMSR_JAX:007900 | Referred to as ScxLin<sup>DTR</sup> in manuscript |
| Genetic reagent (*Mus musculus*) | B6.Cg-Gt(ROSA)26Sortm9 (CAG-tdTomato)Hze/J (*ROSA-Ai9*) | Jackson Laboratory | Stock #: 007909 RRID:IMSR_JAX:007909 | Referred to as ScxLin<sup>Ai9</sup> or ScxLin<sup>Ai9DTR</sup> in manuscript |
| Antibody | Anti-SCXA (rabbit polyclonal) | Abcam | Catalog #: ab58655 RRID:AB_882467 | (1:500) |
| Antibody | Anti-S100a4 (rabbit monoclonal) | Abcam | Catalog #: ab197896 RRID:AB_2728774 | (1:2000) |
| Antibody | Anti-cleaved caspase 3 (rabbit polyclonal) | Cell Signalling Technology | Catalog #: 9661 RRID:AB_2341188 | (1:100) |
| Antibody | Anti-PCNA (mouse monoclonal) | Abcam | Catalog #: ab29 RRID:AB_303394 | (1:100) |
| Antibody | Anti-F4/80 (rabbit polyclonal) | Santa Cruz Biotechnology | Catalog #: sc-26643 RRID:AB_2098331 | (1:500) |
| Antibody | Anti-THBS4 (rabbit monoclonal) | Abcam | Catalog #: ab263898 | (1:250) |
| Antibody | Anti-MFAP5 (rabbit monoclonal) | Abcam | Catalog #: ab203828 | (1:2000) |
| Antibody | Anti-Decorin (Rabbit polyclonal) | Abcam | Catalog #: ab175404 | (1:250) |
| Antibody | Anti-alpha-SMA-Cy3 (mouse monoclonal) | Sigma-Aldrich | Catalog #: C6198 RRID:AB_476856 | (1:200) |
| Antibody | Anti-alpha-SMA-FITC (mouse monoclonal) | Sigma-Aldrich | Catalog #: F3777 RRID:AB_476977 | (1:500) |
| Antibody | Rhodamine Red-X (RRX) AffiniPure F(ab')$_2$ Fragment Donkey Anti-Rabbit IgG (H+L) (Donkey polyclonal) | Jackson ImmunoResearch | Catalog #: 711-296-152 | (1:200) |
| Antibody | Alexa Fluor 488 AffiniPure F(ab')$_2$ Fragment Donkey Anti-Goat IgG (H+L) (Donkey polyclonal) | Jackson ImmunoResearch | Catalog #: 705-546-147 RRID:AB_2340430 | (1:200) |
| Antibody | Rhodamine Red-X (RRX) AffiniPure F(ab')$_2$ Fragment Donkey Anti-Mouse IgG (H+L) (Donkey polyclonal) | Jackson ImmunoResearch | Catalog #: 715-296-150 RRID:AB_2340834 | (1:200) |
| Chemical Compound, Drug | Diphtheria Toxin (DT) | Millipore Sigma | D0564-1MG | 20 ng DT / injection |
| Software, algorithms | GraphPad Prism software | GraphPad Prism (https://graphpad.com) | Version 7.02 | |
| Software, algorithms | OlyVIA software | Olympus (https://www.olympus-lifescience.com/en/support/downloads/) | Version 2.9 | |
| Software, algorithms | ImageJ software | ImageJ (http://imagej.nih.gov/ij/) | | |

## Mice

Scx-Cre mice were generously provided by Dr. Ronen Schweitzer. ROSA-Ai9<sup>F/F</sup> (#007909), and Rosa-DTR<sup>LSL</sup> (#007900) mice were obtained from the Jackson Laboratory (Bar Harbor, ME, USA). ROSA-Ai9<sup>LSL</sup> mice express Tomato red fluorescence in the presence of Cre-mediated recombination (*Madisen et al., 2010*). Scx-Cre mice were crossed to ROSA-Ai9<sup>F/F</sup> mice to label *Scx* lineage cells (Scx<sup>Lin</sup>). Diphtheria toxin receptor (DTR<sup>LSL</sup>) mice can be utilized to temporally ablate cell populations

driven by a non-inducible Cre driver (*Buch et al., 2005*). In short, expression of the diphtheria toxin receptor is inhibited prior to Cre-mediated recombination due to the presence of a STOP cassette flanked by loxp site (Loxp-STOP-Loxp; LSL). Following Cre-mediated recombination the STOP cassette is deleted, resulting in expression of DT receptor, in this case specifically on Scx$^{Lin}$ cells. As such, administration of diphtheria toxin (DT) to these mice results in targeted cell death of Scx$^{Lin}$ cells. Scx-Cre mice were crossed to DTR$^{LSL}$ animals to generate a model of Scx$^{Lin}$ tendon cell depletion (ScxLin$^{DTR}$) and Scx-Cre-; DTR$^{LSL}$ littermates were used as wild-type (WT) controls. To simultaneously deplete and visualize Scx$^{Lin}$ tendon cells, Scx-Cre; Ai9$^{F/F}$ mice were crossed to the DTR$^{LSL}$ to generate a model of Scx-Cre$^{+}$; Ai9$^{F/+}$; DTR$^{F/+}$ littermates (ScxLin$^{Ai9DTR}$) and Scx-Cre$^{+}$; Ai9$^{F/+}$; DTR$^{+/+}$ (ScxLin$^{Ai9}$)were used as WT to visualize ScxLin$^{Ai9}$ cells without depletion. All mouse studies were performed with 10–12 week-old male and female mice except where otherwise noted (*Figure 7* 3–4-week-old male and female mice). All mouse work (injections, surgeries, harvests) were performed in the morning. Mice were kept in a 12 hr light/dark cycle.

## Flexor tendon repair

Complete transection and repair of the murine flexor digitorum longus (FDL) tendon was performed as previously described (*Ackerman and Loiselle, 2016*). Mice received a 15–20 μg injection of sustained-release buprenorphine. Mice were anesthetized with Ketamine (60 mg/kg) and Xylazine (4 mg/kg). To reduce chances of rupture at the repair site, the FDL tendon was first transected at the myotendinous junction and the skin was closed with a 5–0 suture. This MTJ transection results in a transient decrease in tendon loading, with progressive reintegration of the MTJ observed by D7-10 post-surgery. As shown by the absence of αSMA staining in the uninjured tendon adjacent to the repair (*Figure 5—figure supplement 1*), this transient alteration in loading does not induce a widespread tendon response such as degeneration, remodeling, or cellular activation. Next, a small incision was made to the posterior surface of the right hind paw, the FDL tendon was isolated from surrounding tissue and completely transected. The tendon was repaired using 8–0 suture and the skin was closed with a 5–0 suture. Animals resumed prior cage activity, food intake, and water consumption immediately following surgery.

## Quantification of tendon cell depletion and Scx$^{Lin}$cells

ScxLin$^{DTR}$ mice were injected with 20 ng of diphtheria toxin (DT) for five consecutive days (100 ng total DT). Uninjured hind paws were harvested 10 days after the final injection for frozen sectioning. Scx$^{Lin}$ hind paws were harvested uninjured hind paws were fixed in 10% NBF for 24 hr, decalcified in 14% EDTA for four days, and processed in 30% sucrose to cryo-protect the tissue. Samples were embedded using Cryomatrix (Thermo Fisher Scientific, Waltham, MA, USA) and sectioned into 8 mm sagittal sections using an established cryotape-transfer method (*Dyment et al., 2016*). Sections were stained with DAPI to visualize nuclei and imaged using a VS120 Virtual Slide Microscope (Olympus, Waltham, MA). Using Image J, a region of interest (ROI) was drawn around the tendon and an area was obtained. For ScxLin$^{DTR}$ and WT littermate mice, nuclei within the ROI were manually counted and total nuclei number was normalized to area. For Scx$^{Lin}$ mice, fluorescent cells in uninjured sections were manually counted and Scx$^{Lin+}$ cells reported as a percentage of total cells counted in each section. An n = 3–4 (adult mice) or n = 3–5 (young mice) was used for quantification.

## Paraffin histology and immunofluorescence

ScxLin$^{DTR}$ hind paws were harvested 10 days after the final DT injection for homeostasis studies, and at 14- and 28 days post-repair. Additionally, uninjured ScxLin$^{DTR}$ hind paws from adult (10–12 weeks) and pre-pubescent (3–4 weeks) mice were harvested 3 months following the final DT injection to assess effects of tendon cell depletion on tendon growth and homeostasis. Hind paws were fixed in 10% neutral buffered formalin (NBF) at room temperature for 72 hr and were subsequently decalcified in Webb Jee EDTA (pH 7.2–7.4) for 7 days at room temperature, processed, and embedded in paraffin. Three-micron sagittal sections were utilized for analysis, except for the 3 month study which were cut at 5-μm to facilitate SHG imaging. ScxLin$^{DTR}$ repair sections were stained with Alcian blue/hematoxylin and Orange G (ABHOG) or Hematoxylin and eosin (H and E) to assess tissue morphology and cellularity, and Masson's Trichrome to assess collagen deposition. For immunofluorescent

staining, sections were stained with Cleaved Caspase 3 (1:100, Cat#: 9661, Cell Signaling, Danvers, MA), PCNA (1:100, Cat#:ab29, Abcam, Cambridge, MA), F4/80 (1:500, Cat#: sc-26643, Santa Cruz, Dallas, TX), α-SMA-CY3 (1:200, Cat#: C6198, Sigma Life Sciences, St. Louis, MO), α-SMA-FITC (1:500, Cat#: F3777, Sigma Life Sciences, St. Louis, MO) S100a4 (1:2000, Cat#: ab197896, Abcam, Cambridge, MA), SCXA (1:500, ab58655, Abcam, Cambridge, MA), Decorin (1:250, Cat # ab175404, Abcam), Thbs4 (1:250, Cat # ab263898, Abcam), Mfap5 (1: 2000, Cat # ab203828, Abcam), or tdTomato (1:500, Cat#: AB8181, SICGEN, Cantanhede, Portugal). Sections were counterstained with the nuclear DAPI stain and imaged with a VS120 Virtual Slide Microscope (Olympus, Waltham, MA).

## Quantification of fluorescence

Fluorescent images scanned by the virtual slide scanner were quantified using Visiopharm image analysis software v.6.7.0.2590 (Visiopharm, Hørsholm, Denmark). Automatic segmentation via a threshold classifier was utilized to define and quantify specific cell populations based on fluorescence. An ROI was drawn to encapsulate both the scar tissue and tendon stubs. The area of fluorescent signal was determined and normalized to the total ROI area to determine percentages of each cell type. An n = 4 was used for quantification. Three samples were utilized for quantification of *Scx* + and *S100a4*+ cells in uninjured ScxLin$^{DTR}$ hind paws 10 days after the final DT injection. Fluorescent cells were manually counted within an ROI and normalized to either the ROI area or to both the ROI area and total cell number.

## Quantitative assessment of gliding function and biomechanical properties

Gliding function of uninjured and repaired ScxLin$^{DTR}$ tendons was assessed as previously described (*Hasslund et al., 2008*). Hindlimbs were harvested at the knee-joint and the FDL tendon was disconnected at the myotendinous junction. The FDL tendon was secured between two pieces of tape using superglue and the tendon was loaded incrementally with small weights ranging from 0 to 19 g. Images were captured unloaded and after each load and measurements of the flexion angle of the metatarsophalangeal (MTP) joint were made using Image J. Gliding resistance was derived from the changes in MTP flexion angle over the range of applied loads. An increase in Gliding Resistance and decrease in MTP Flexion Angle is associated with restricted range of motion and increased scar tissue. After conclusion of gliding testing, the FDL tendon was released from the tarsal tunnel while the proximal end of the tendon and the toes of the hind paw were secured into an Instron 8841 uni-axial testing system (Instron Corporation, Norwood, MA). The tendon was loaded until failure at a rate of 30 mm/minute. Seven to 12 samples per genotype per time-point were assessed.

## RNA extraction, next-generation sequencing, and data analysis for RNA-Seq

Tendons (three samples per genotype per time point) were harvested at 14- and 28 days post-repair and flash frozen in liquid nitrogen. Total RNA was isolated using the Bullet Blender (Next Advance) to homogenize the tissue. The RNA was isolated from the resulting extract using Trizol (Life Technologies, Carlsbad, CA) and the RNeasy Plus Micro Kit (Qiagen, Valencia, CA) per manufacturer's recommendations. The total RNA concentration was determined with the NanoDrop 1000 spectrophotometer (NanoDrop, Wilmington, DE) and RNA quality assessed with the Agilent Bioanalyzer (Agilent, Santa Clara, CA). The RNA integrity number (RIN) for all harvested samples was 8.4 ± 0.85 (mean ± standard deviation). The TruSeq Stranded mRNA Sample Preparation Kit (Illumina, San Diego, CA) was used for next-generation sequencing library construction per manufacturer's protocols. Briefly, mRNA was purified from 200 ng total RNA with oligo-dT magnetic beads and fragmented. First-strand cDNA synthesis was performed with random hexamer priming followed by second-strand cDNA synthesis using dUTP incorporation for strand marking. End repair and 3′ adenylation was then performed on the double-stranded cDNA. Illumina adaptors were ligated to both ends of the cDNA and amplified with PCR primers specific to the adaptor sequences to generate cDNA amplicons of approximately 200–500 bp in size. The amplified libraries were hybridized to the Illumina flow cell and single end reads were generated for each sample using Illumina Nova-Seq6000. The generated reads were demultiplexed using bcl2fastq version 2.19.0. Data cleaning and quality control was accomplished using FastP version 0.20.0. Read quantification was

accomplished using subread-1.6.4 package (featureCounts). Data normalization and differential expression analysis of ScxLin$^{DTR}$ relative to WT at a given time point was performed using DESeq2-1.22.1 with an adjusted p-value threshold of 0.05 on each set of raw expression measures. The 'lfcShrink' method was applied, which moderates log2 fold-changes for lowly expressed genes. DeSeq2 data was uploaded to Qiagen's ingenuity pathway analysis (IPA, http://www.ingenuity.com) and submitted to core analysis. Canonical pathways, upstream regulators, and downstream disease and functions were deemed biologically and statistically significant when the calculated ABS(z-score) >2 and p-value<0.05. Additionally, imposed upon the upstream regulator data was an expression log ratio cut-off of 0.5, and agreement between predicted activation state and directionality of expression log ratio. The data consists of three biological replicates per genotype per timepoint. The data generated in this study have been uploaded to the Gene Expression Omnibus under accession number GSE156157.

## Second harmonic generation two-photon confocal imaging

Five-micron paraffin sections of WT and ScxLin$^{DTR}$ hind paws were utilized for second harmonic generation (SHG) imaging. Sections were scanned with a Spectra-Physics MaiTai HP DeepSee Ti:Sapphire Laser, tuned to 1000 nm, under 25x magnification, with a 2.5X optical zoom, with a step size of 0.25 μm. 3D projections of image stacks were generated using the 3D-Project macro in ImageJ and analyzed for collagen fibril uniformity using the Directionality macro. The Directionality macro utilizes Fourier transform analysis to derive spatial orientation of image stacks. Three to 10 samples per genotype were used to quantify overall collagen fibril dispersion for pre-pubescent studies, while sections were analyzed from 7 to 8 mice per genotype for the adult homeostasis studies. To quantify spatial dispersion in adult samples, each image stack was divided into equal thirds (top, middle, and bottom), and dispersion was calculated within each region using Directionality macro as above. N = 6 per genotype per age group were used to quantify the spatial collagen fibril dispersion.

## Transmission electron microscopy imaging and analysis

FDL tendons were isolated (N = 4 for WT; N = 3 for ScxLin$^{DTR,10weeks}$) and fixed in Glutaraldehyde Sodium Cacodylate fixative. One-micron axial sections were cut and stained with Toluidine blue. One-micron sections were then trimmed to 70 nm and stained with uranyl acetate and lead citrate. Sections were placed on grids for imaging on a Hitachi 7650 Analytical TEM. Three non-overlapping images were taken from mid-substance of each tendon at ×40,000 magnification. For measurement of fibril diameter, a region of interest (ROI) was determined within each image so that a minimum of 80 fibrils could be measured. Diameters were measured along the y-axis. The perimeter and the area of the collagen fibrils were quantified. The radii based on the calculated perimeter and area were quantified. The ratio of these two radii represent a measure of fibril roundness (fibril irregularity factor; FIF). An FIF different than one suggest that the fibril is not a perfect circle.

## Statistical analysis and animal stratification

Experimental N determined based on previously published work (*Best and Loiselle, 2019*; *Best et al., 2019*). Quantitative data was analyzed via GraphPad Prism and is presented as mean ± standard error of the mean (SEM). Either a student's t-test or two-way analysis of variance (ANOVA) with Sidak's multiple comparisons test was used to analyze data when data was normal. A Mann-Whitney test was utilized when data was not distributed normally [*Scx*+ cells normalized to area and total cell number (*Figure 1—figure supplement 1C*), D28 F4/80 and S100a4 immunofluorescence (*Figure 4*)]. GraphPad Prism was used to detect outlier data points (ROUT method, Q-value = 1%) and no outliers were found. Mice were randomly selected for specific experimental outcome metrics prior to surgery and quantitative data (ex. fluorescence quantification, gliding, and biomechanical properties) were analyzed in a blinded manner. For all experiments, an N = 1 represents one mouse. p values ≤ 0.05 were considered significant. * indicates p<0.05, ** indicates p<0.01, *** indicates p<0.001, **** indicates p<0.0001.

## Study approval

This study was carried out in strict accordance with the recommendations in the Guide for the Care and Use of Laboratory Animals of the National Institutes of Health. All animal procedures described were approved by the University Committee on Animal Research (UCAR) at the University of Rochester Medical Center.

## Acknowledgements

We thank the Histology, Biochemistry and Molecular Imaging (HBMI) and the Biomechanics, Biomaterials and Multimodal Tissue Imaging (BBMTI) for technical assistance with the histology and biomechanical testing, respectively. We would also like to thank the URMC Multiphoton and Analytical Imaging Center (MAGIC) for assistance with Second Harmonic Generation Imaging, the UR Genomics Research Core for assistance with RNA sequencing experiment and the Electron Microscopy Core for assistance with the transmission electron microscopy data.

## Additional information

### Funding

| Funder | Grant reference number | Author |
|---|---|---|
| National Institute of Arthritis and Musculoskeletal and Skin Diseases | F31 AR074815 | Katherine T Best |
| National Institute of Arthritis and Musculoskeletal and Skin Diseases | K01AR068386 | Alayna E Loiselle |
| National Institute of Arthritis and Musculoskeletal and Skin Diseases | R01AR073169 | Alayna E Loiselle |
| National Institute of Arthritis and Musculoskeletal and Skin Diseases | R01AR070765 | Mark R Buckley |
| National Institute of Arthritis and Musculoskeletal and Skin Diseases | T32 AR076950 | Anne EC Nichols |

The funders had no role in study design, data collection and interpretation, or the decision to submit the work for publication.

### Author contributions

Katherine T Best, Conceptualization, Formal analysis, Funding acquisition, Methodology, Writing - original draft, Writing - review and editing; Antonion Korcari, Conceptualization, Formal analysis, Methodology, Writing - original draft, Writing - review and editing; Keshia E Mora, Anne EC Nichols, Formal analysis, Methodology, Writing - review and editing; Samantha N Muscat, Formal analysis, Writing - review and editing; Emma Knapp, Data curation, Writing - review and editing; Mark R Buckley, Supervision, Methodology, Writing - review and editing; Alayna E Loiselle, Conceptualization, Supervision, Funding acquisition, Writing - original draft, Writing - review and editing

### Author ORCIDs

Alayna E Loiselle (iD) https://orcid.org/0000-0002-7548-6653

### Ethics

Animal experimentation: This study was performed in strict accordance with the recommendations in the Guide for the Care and Use of Laboratory Animals of the National Institutes of Health. All of the animals were handled according to approval by the University Committee on Animal Resources (UCAR) for protocols #2014-004E and 2017-030 at the University of Rochester. All surgery was performed under ketamine anesthesia, and every effort was made to minimize suffering.

## Decision letter and Author response

Decision letter https://doi.org/10.7554/eLife.62203.sa1
Author response https://doi.org/10.7554/eLife.62203.sa2

## Additional files

### Supplementary files

• Transparent reporting form

### Data availability

Sequencing data have been deposited in GEO under accession code GSE156157. All other data generated during this study are included in the manuscript and supporting files.

The following dataset was generated:

| Author(s) | Year | Dataset title | Dataset URL | Database and Identifier |
|---|---|---|---|---|
| Best KT, Loiselle AE | 2021 | RNA-seq analysis of Scx-lineage cell depletion to investigate tendon cell functions during flexor tendon healing | https://www.ncbi.nlm.nih.gov/geo/query/acc.cgi?acc=GSE156157 | NCBI Gene Expression Omnibus, GSE156157 |

The following previously published datasets were used:

| Author(s) | Year | Dataset title | Dataset URL | Database and Identifier |
|---|---|---|---|---|
| Lincoln J, Barnette DN | 2014 | Expression data from embryonic day 15.5 atrioventricular canal regions were isolated from Scx-/- and Scx+/+ mice | https://www.ncbi.nlm.nih.gov/geo/query/acc.cgi?acc=GSE57423 | NCBI Gene Expression Omnibus, GSE57423 |
| Mendias C, Swanson J | 2019 | Single cell transcriptional atlas of mouse Achilles tendons | https://www.ncbi.nlm.nih.gov/geo/query/acc.cgi?acc=GSE138515 | NCBI Gene Expression Omnibus, GSE138515 |

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
