## [Decision Letter]

**Acceptance summary:**

The study utilized a mouse model of Scx^Lin^ cell depletion to directly assess the function of tendon cells during tendon healing. The finding that Scx^Lin^-depleted tendon cells result in improved biomechanical properties is a noteworthy observation. This is buttressed by RNA sequencing data that provides useful mechanistic information. Issues raised by the reviewers have been adequately addressed.

**Decision letter after peer review:**

Thank you for submitting your article "Scleraxis-Lineage Cell Depletion Improves Tendon Healing and Disrupts Adult Tendon Homeostasis" for consideration by *eLife*. Your article has been reviewed by three peer reviewers, and the evaluation has been overseen by Mone Zaidi as the Reviewing Editor and Clifford Rosen as the Senior Editor. The following individuals involved in review of your submission have agreed to reveal their identity: Nathaniel Dyment (Reviewer #1); Stavros Thomopoulos (Reviewer #3).

The reviewers have discussed the reviews with one another and the Reviewing Editor has drafted this decision to help you prepare a revised submission.

This is an interesting study exploring the role of Scx^Lin^ cells on tendon healing, post-natal development, and adult homeostasis. The findings are considered worthy and potentially publishable, particularly due to the unexpected result of an absent phenotype in haploinsufficiency or Scx lineage cells. With that said, several aspects need to be strengthened with new experiments. While point 4 is considered important as it would enhance the conclusions, the Editors are cognizant of the time that may be required. Hence, the latter aspect could be discussed.

Essential revisions

1) In addition to tenocytes, ScxCre will also label epitenon cells and many other cells adjacent to tendons including neighboring sheath cells (see Guak Kim Tan's *eLife* paper, Jan 2020). Since these cells likely also contribute to the healing response, ablation of these cells may be one cause of improved healing. It is critical that Scx^lin^ cells are assessed and quantified during the healing process. The authors should use the RosaT reporter and/or ScxGFP reporter to perform lineage tracing of ScxCre cells after ablation and/or visualization of ScxGFP cells.

2) Since the authors reported grip-to-grip structural properties and not local strains or material properties, the reported structural properties will be impacted by changes to the adjacent tendon stubs in addition to the healing tissue. Since the MTJ is cut, presumably there will be an unloading-induced remodeling response that may be impacted in the DTA mice because 57% of the cells were ablated. Instead of redoing the biomechanics with local strain measurements and material properties, the authors should analyze the adjacent tendon stubs (i.e., tendon midsubstance adjacent to injury site) using using aSMA staining, SHG and/or TEM (with fibril diameter/distribution quantification) to determine if that tissue was protected from unloading-induced degradation.

3) Since the injury site is a heterogeneous mix of cells, the authors should validate and localize matrix markers identified from RNASeq using immunohistochemistry and/or in situ hybridization. These assays would give insight into the mechanism by which the ablated tendons show improved healing.

4) The reviewers prefer a delayed ablation experiment. They also suggest that the authors inject dead cells prior to injury to see if the effects are not due to Scx cell ablation per se, but instead to induction of an immune response that leads to better healing. While these experiments would exceed the allowable 6 month time for revision, the authors should discuss these points. For instance, Vagnozzi et al., 2019 found that in a cardiac wound injury-repair model, improved healing associated with cell therapies was due to the inflammatory response and recruitment of macrophages (not injection of the cells itself).

---

## [Author Response]

Essential revisions1) In addition to tenocytes, ScxCre will also label epitenon cells and many other cells adjacent to tendons including neighboring sheath cells (see Guak Kim Tan's eLife paper, Jan 2020). Since these cells likely also contribute to the healing response, ablation of these cells may be one cause of improved healing. It is critical that Scx^lin^ cells are assessed and quantified during the healing process. The authors should use the RosaT reporter and/or ScxGFP reporter to perform lineage tracing of ScxCre cells after ablation and/or visualization of ScxGFP cells.

This is a very good point. To address this, we have generated Scx-Cre+; Rosa-Ai9; Rosa-DTR^LSL^ mice to track changes in Scx^LinAi9^ content before injury and during healing, relative to Scx-Cre+; Rosa-Ai9; DTR^WT^ controls. In new Figure 1 we show that there is a significant decrease in ScxLin^Ai9^ cells in the depleted tendons through 38 days post-depletion. Interestingly, no changes in ScxLin^Ai9^ cells were observed in depleted repairs at D14 post-surgery. However, at D28 a significant decrease in ScxLin^Ai9^ cells were observed, relative to WT repairs (new Figure 4). We have also added a schematic to this figure to help with interpretation of these surprising data and have discussed this at length in the Discussion. Briefly, we think that these data suggest that the Scx^Lin^ cells that are present in the adult tendon during homeostasis make their predominant contribution to healing at or around D28. Thus, the effects of depletion are manifested at this time as evidenced by changes in ScxLin^Ai9^ content, function, and transcriptional profile. In contrast, we hypothesize that the lack of differences in ScxLin^Ai9^ content, function and transcriptional profile at D14 are due to a potential increase in cells that express *Scx* in response to injury. These cells would still be labelled as ScxLin^Ai9^ due to the non-inducible nature of the Scx-Cre driver used, but would not have been targeted for depletion due to their lack of *Scx* expression prior to injury. The induction of *Scx* expression after injury in the tendon is well supported by prior studies (Dyment et al., 2014, Dyment et al., PlosOne 2013, Howell et al., 2017).

2) Since the authors reported grip-to-grip structural properties and not local strains or material properties, the reported structural properties will be impacted by changes to the adjacent tendon stubs in addition to the healing tissue. Since the MTJ is cut, presumably there will be an unloading-induced remodeling response that may be impacted in the DTA mice because 57% of the cells were ablated. Instead of redoing the biomechanics with local strain measurements and material properties, the authors should analyze the adjacent tendon stubs (i.e., tendon midsubstance adjacent to injury site) using using aSMA staining, SHG and/or TEM (with fibril diameter/distribution quantification) to determine if that tissue was protected from unloading-induced degradation.

Thank you for this question. The MTJ transection to protect the initial repair from rupturing results in only a transient decrease in tendon loading. By 7-10 days post-surgery, there is substantial re-integration of the MTJ, however, we do acknowledge that this altered loading environment could impact the un-injured tendon adjacent to the repair site. To address this, we examined aSMA staining in the uninjured sections of the tendon proximal/ distal to the repair site (Figure 5—figure supplement 1). A complete lack of aSMA staining was observed in these sections of the tendon, with aSMA staining concentrated at the repair site, suggesting that the transient decrease in tendon loading due to MTJ transection does not result in a widespread tendon response unloading-induced degeneration, remodeling or cellular response. We have also edited the Materials and methods section to clarify the transient nature of the MTJ-transection induced alterations in loading.

3) Since the injury site is a heterogeneous mix of cells, the authors should validate and localize matrix markers identified from RNASeq using immunohistochemistry and/or in situ hybridization. These assays would give insight into the mechanism by which the ablated tendons show improved healing.

To validate and provide spatial information related to the RNAseq study, we performed immunofluorescence for Decorin (Dcn), Thrombospondin 4 (Thbs4) and Microfibril Associated Protein 5 (Mfap5) ECM components that were differentially regulated between WT and ScxLin^DTR^ repairs at D28 post-surgery (Figure 7—figure supplement 1). Consistent with the RNAseq data, we observe substantial increases in the intensity and extent of staining of these ECM components in ScxLin^DTR^ vs. WT repairs, further supporting an enhanced matrix response in ScxLin^DTR^ repairs that differs from that of WT. These data are consistent with an enrichment for “Fibrosis” related genes in ScxLin^DTR^ repairs, though it is clear that these alterations in matrix composition and quantity are actually more reflective of enhanced healing rather than fibrosis.

4) The reviewers prefer a delayed ablation experiment. They also suggest that the authors inject dead cells prior to injury to see if the effects are not due to Scx cell ablation per se, but instead to induction of an immune response that leads to better healing. While these experiments would exceed the allowable 6 month time for revision, the authors should discuss these points. For instance, Vagnozzi et al., 2019 found that in a cardiac wound injury-repair model, improved healing associated with cell therapies was due to the inflammatory response and recruitment of macrophages (not injection of the cells itself).

This is a very good suggestion and is the focus of on-going studies in our laboratory. However, we have confined the current studies to depletion of ScxLin cells prior to injury due in large part to the potentially confounding effects of depleting Scx^Lin^ during healing. Since systemic treatment with DT is lethal to Scx-Cre+; DTR^LSL^ animals, likely due to *Scx* expression in the heart and other organs, we have utilized local injection of DT directly into the hindpaw and tendon. We have conducted pilot experiments and determined that local injections, even of saline, during the healing process results in some alterations in healing. As such, alternative approaches to locally deliver DT during healing are being developed. Therefore, it is not feasible to conduct the proposed experiments at this time. Moreover, as supported by the new data in Figure 4, there are very likely cells that turn-on Scx expression in response to injury, and the overlap between these new Scx^Lin^ cells and those from the adult tendon prior to injury is unknown. As such, it is likely that a delayed ablation approach would target different subpopulations of Scx^Lin^ cells and our goal with this study was to understand the functional contribution of Scx^Lin^ cells that reside in the adult tendon prior to injury. We have added text on this to the Discussion.

Regarding the potential effects of cell death on the immune response, as shown in Figure 1—figure supplement 3, we did not detect any inflammatory infiltration following Scx^Lin^ cell depletion. Moreover, as shown in Figure 5, no changes in overall macrophage content were observed between depleted and WT controls at 14 or 28 days post-surgery. Finally, alterations in the immune response was not noted as a primary pathway/ process that was altered in the RNAseq experiment. We have expanded on this in the Discussion.